# PAGAR: Taming Reward Misalignment in Inverse Reinforcement Learning-Based Imitation Learning with Protagonist Antagonist Guided Adversarial Reward

## Abstract

Many imitation learning (IL) algorithms employ inverse reinforcement learning (IRL) to infer the underlying reward function that an expert is implicitly optimizing for, based on their demonstrated behaviors. However, a misalignment between the inferred reward and the true task objective can result in task failures. In this paper, we introduce Protagonist Antagonist Guided Adversarial Reward (PAGAR), a semi-supervised reward design paradigm to tackle this reward misalignment problem in IRL-based IL. We identify the conditions on the candidate reward functions under which PAGAR can guarantee to induce a policy that succeeds in the underlying task. Furthermore, we present a practical on-and-off policy approach to implement PAGAR in IRL-based IL. Experimental results show that our algorithm outperforms competitive baselines on complex IL tasks and zero-shot IL tasks in transfer environments with limited demonstrations.

## 1 Introduction

The central principle of reinforcement learning (RL) is reward maximization Mnih et al. (2015); Silver et al. (2016); Bertsekas (2009). The effectiveness of RL thus hinges on having a proper reward function that drives the desired behaviors Silver et al. (2021). Inverse reinforcement learning (IRL) Ng & Russell (2000); Finn et al. (2017) is a well-known approach that aims to learn an agent's reward from a set of candidate rewards by observing its behaviors. IRL is also often leveraged as a subroutine in imitation learning (IL) where the learned reward function is used to train a policy via RL Abbeel & Ng (2004); Ho & Ermon (2016). However, it is challenging for IRL to identify a proxy reward function that is aligned with the true task objective. One common cause of reward misalignment in IRL-based IL is reward ambiguity – multiple reward functions can be consistent with expert demonstrations, even in the limit of infinite-data Ng & Russell (2000); Cao et al. (2021); Skalse et al. (2022a;b); Skalse & Abate (2022), but only some of those reward functions may be aligned with the task objective. Training a policy with a misaligned reward can result in reward hacking and task failures Hadfield-Menell et al. (2017); Amodei et al. (2016); Pan et al. (2022).

Our key insight into the problem of reward misalignment in IRL-based IL is that *there is a disconnect between the principle of reward maximization and the notion of task success or failure – reward maximization is often neither a sufficient nor necessary condition for accomplishing the underlying task*. We consider the notion of task success and failure as a mapping from policies to binary outcomes, i.e., $\Phi(\pi) \in \{\texttt{true}, \texttt{false}\}$ where $\Phi(\pi) = \texttt{true}$ meaning $\pi$ succeeds in the task and otherwise $\pi$ fails the task. We propose the following definition for identifying whether a reward function is aligned with the given task.

**Definition 1 (Task-Reward Alignment).** Let $U_r(\pi) \in \mathbb{R}$ measure the performance of any policy $\pi \in \Pi$ from a policy space $\Pi$ under a reward function $r$, and $\mathbb{U}_r = [\min_\pi U_r(\pi), \max_\pi U_r(\pi)]$ be the range of $U_r(\pi)$ for $\pi \in \Pi$. A reward function $r$ is said to be **aligned** with the task if and only if there exists a non-empty success interval $S_r \subset \mathbb{U}_r$ and a failure interval $F_r \in \mathbb{U}_r$ such that (1) $sup\, S_r = \max_\pi U_r(\pi) \wedge \inf S_r \leq \sup S_r$, (2) $F_r = \emptyset \vee (\inf F_r = \min_\pi U_r(\pi) \wedge \sup F_r < \inf S_r)$,

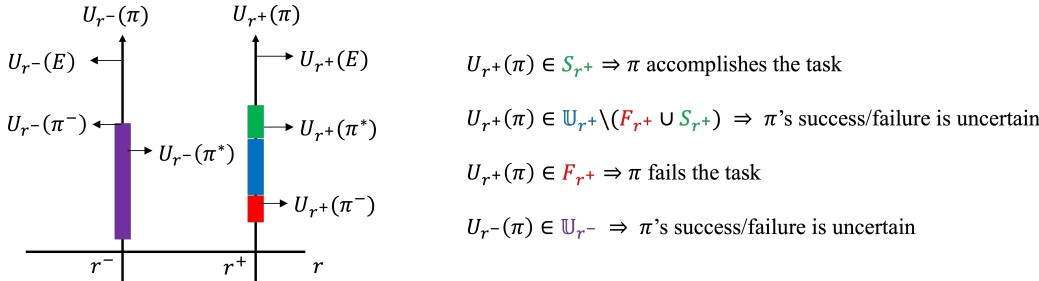

Figure 1: The $r$-axis indicates different reward functions. The reward function $r^-$ is misaligned with the task, whereas $r^+$ is aligned with the task. The two vertical lines indicate the values of $U_{r^+}(\pi)$ and $U_{r^-}(\pi)$. The purple bar indicates $\mathbb{U}_{r^-}$, the interval of $U_{r^-}(\pi)$. The green, blue, and red bars jointly indicate $\mathbb{U}_{r^+}$, the interval of $U_{r^+}(\pi)$. If a policy's performance is in the green bar interval $S_{r^+}$, this policy is guaranteed to accomplish the task; if a policy's performance is in the red bar interval $F_{r^+}$, this policy is guaranteed to fail the task; otherwise, the policy's success or failure in the task is uncertain. The performance margin between the expert demonstrations $E$ and any policy $\pi$ measured under a reward function $r$ is $U_r(E) - U_r(\pi)$. The reward function $r^-$ is the optimal solution for IRL while $r^+$ is sub-optimal, i.e., $U_{r^-}(E) - \max_\pi U_{r^-}(\pi) > U_{r^+}(E) - \max_\pi U_{r^+}(\pi)$. As a result, IRL-based IL learns a policy $\pi^-$, which is optimal under $r^-$ but achieves a low performance $U_{r^+}(\pi^-) \in F_{r^+}$ under $r^+$, thus failing the task. We propose to learn $\pi^*$ that achieves high $U_{r^+}(\pi^*)$ and $U_{r^-}(\pi^-)$. The idea is that even though $\pi^*$ may not be optimal under $r^+$ or $r^-$, the performance margins from $E$ to $\pi^*$, i.e., $U_{r^+}(E) - U_{r^+}(\pi^*)$ and $U_{r^-}(E) - U_{r^-}(\pi^*)$, are both small.

(3) for any policy $\pi \in \Pi$, $U_r(\pi) \in S_r \Rightarrow \Phi(\pi)$ and $U_r(\pi) \in F_r \Rightarrow \neg\Phi(\pi)$. If $S_r$ and $F_r$ do not exist, $r$ is said to be **misaligned**.

The definition states that if a reward function $r$ is aligned with the task, then a policy $\pi$ can accomplish the task as long as it can achieve a high enough $U_r(\pi)$ such that $U_r(\pi) \in S_r$. On the other hand, if $r$ is misaligned with the task, then even its optimal policy may not accomplish the task. During the iterative process of IRL-based IL, multiple candidate reward functions are inferred from the demonstrations. Our idea is to train a policy to achieve high performance under a selected subset of those reward functions, even when some reward functions may not be the optimal solution of IRL. The intuition is to gain from those reward functions a collective validation of the policy's similarity with respect to the expert demonstrations. We illustrate this idea with Figure 1 where IRL infers an optimal reward function $r^-$ and a sub-optimal reward function $r^+$ from the expert demonstrations $E$. Suppose that $r^+$ is aligned with the underlying task while $r^-$ is misaligned. IRL-based IL will use the optimal solution $r^-$ to learn a policy $\pi^-$ which performs optimally under $r^-$ but much worse under $r^+$, i.e., $U_{r^+}(\pi^-) \in F_{r^+}$, thus failing the task. Our goal is to learn a policy $\pi^*$ to attain high performance under both $r^+$ and $r^-$ without explicitly identifying which reward function is aligned with the task. Even though $\pi^*$ may not be optimal under either $r^+$ or $r^-$, $\pi^*$ has a small performance margin with respect to $E$ under both $r^+$ and $r^-$.

We concretize our proposition in a novel semi-supervised reward design paradigm called Protagonist Antagonist Guided Adversarial Reward (PAGAR). Treating the policy trained for the underlying task as a protagonist, PAGAR adversarially searches for a reward function to challenge the protagonist policy to achieve performances on par with the optimal policy under that reward function (which we call an antagonist policy). Then, by iteratively training the protagonist policy with the searched reward functions, we can mitigate the problem of reward misalignment due to optimizing for a single but misaligned reward. This protagonist and antagonist setup is inspired by the concept of unsupervised environment design (UED) from Dennis et al. (2020). In this paper, we develop novel theories for semi-supervised reward design and prove that PAGAR can mitigate the problem of reward misalignment in IRL-based IL. In addition, we propose an on-and-off-policy approach to implementing PAGAR-based IL. We summarize our contributions below.

- We propose a semi-supervised reward design paradigm for mitigating reward misalignment. We identify the technical conditions on the candidate reward functions for avoiding failures and guaranteeing success in the underlying task.

- We develop an on-and-off-policy approach to implementing this paradigm in IL.
- Experimental results demonstrate that our algorithm outperforms competitive baselines on complex IL tasks and zero-shot IL tasks in transfer environments with limited demonstrations.

## 2 RELATED WORKS

Reward misalignment has been studied under various contextx such as reward misspecification Pan et al. (2022); Skalse & Abate (2022), reward hacking Skalse et al. (2022b), and reward ambiguity Skalse & Abate (2022); Ng & Russell (2000); Cao et al. (2021). Previous attempts to ensure reward alignment in RL include using logical structures Toro Icarte et al. (2019); Hasanbeig et al. (2019). In IRL-based IL, the efforts to alleviate the reward ambiguity problem include Max-Entropy IRL from Ziebart et al. (2008), Max-Margin IRL from Abbeel & Ng (2004); Ratliff et al. (2006), and Bayesian IRL from Ramachandran & Amir (2007). However, these approaches do not fundamentally address the reward misalignment problem. Our work shares a viewpoint with Metelli et al. (2021) and Lindner et al. (2022) that reward ambiguity can be circumvented by focusing on the feasible reward set, rather than a single reward function. However, while those works primarily consider the optimal solution set of IRL. GAN-based methods Ho & Ermon (2016); Jeon et al. (2018); Finn et al. (2016); Peng et al. (2019); Fu et al. (2018) leverage the expressivity of neural networks to learn reward functions and policies from limited demonstrations. However, they do not specifically address the reward misalignment problem. Other efforts on resolving reward misalignment in IRL-based IL resort to using additional information other than demonstrations, such as human preferences over trajectories Dorsa Sadigh et al. (2012); Brown et al. (2019), expert behaviors from multiple tasks Shah et al. (2019), logical structures for reward functions Zhou & Li (2022a;b), and formal verification Zhou & Li (2018). In contrast, our approach does not rely on such additional information.

## 3 PRELIMINARIES

**Reinforcement Learning (RL)** models the environment as a Markov Decision process $\mathcal{M} = \langle \mathbb{S}, \mathbb{A}, \mathcal{P}, d_0 \rangle$ where $\mathbb{S}$ is the state space, $\mathbb{A}$ is an action space, $\mathcal{P}(s'|s,a)$ is the probability of reaching a state $s'$ by performing an action $a$ at a state $s$, and $d_0$ is an initial state distribution. A *policy* $\pi(a|s)$ determines the probability of an RL agent performing an action $a$ at state $s$. By successively performing actions for $T$ steps from an initial state $s^{(0)} \sim d_0$, a *trajectory* $\tau = s^{(0)} a^{(0)} s^{(1)} a^{(1)} \ldots s^{(T)}$ is produced. A state-action based *reward function* is a mapping $r : S \times A \to \mathbb{R}$. The soft Q-value function of $\pi$ is $\mathcal{Q}_\pi(s,a) = r(s,a) + \gamma \cdot \mathbb{E}_{s' \sim \mathcal{P}(\cdot|s,a)}[\mathcal{V}_\pi(s')]$ where $\mathcal{V}_\pi$ is the soft state-value function of $\pi$ defined as $\mathcal{V}_\pi(s) := \mathbb{E}_{a \sim \pi(\cdot|s)}[\mathcal{Q}_\pi(s,a)] + \mathcal{H}(\pi(\cdot|s))$, and $\mathcal{H}(\pi(\cdot|s))$ is the entropy of $\pi$ at a state $s$. The soft advantage of performing action $a$ at state $s$ then following a policy $\pi$ afterwards is then $\mathcal{A}_\pi(s,a) = \mathcal{Q}_\pi(s,a) - \mathcal{V}_\pi(s)$. The expected return of $\pi$ under a reward function $r$ is given as $\eta_r(\pi) = \mathbb{E}_{\tau \sim \pi}[\sum_{t=0}^{T} r(s^{(t)}, a^{(t)})]$. With a little abuse of notations, we denote $r(\tau) := \sum_{t=0}^{T} \gamma^t \cdot r(s^{(t)}, a^{(t)})$, and $\mathcal{H}(\pi) := \sum_{t=0}^{T} \mathbb{E}_{s^{(t)} \sim \pi}[\gamma^t \cdot \mathcal{H}(\pi(\cdot|s^{(t)}))]$. The objective of entropy-regularized RL is to learn a policy $\pi$ that maximizes $J_{RL}(\pi; r) = \eta_r(\pi) + \mathcal{H}(\pi)$.

**Inverse Reinforcement Learning (IRL)** assumes that a set $E = \{\tau_1, \ldots, \tau_N\}$ of expert demonstrations is provided instead of the reward function. It is also assumed that the expert demonstrations are sampled from the roll-outs of the expert's policy $\pi_E$. Given a candidate set of reward functions $R$, Maximum Entropy IRL Ziebart et al. (2008) solves for the reward function that maximizes the IRL loss function $J_{IRL}(r) = \max_\pi \eta_r(\pi) + \mathcal{H}(\pi) - \eta_r(\pi_E)$ where $\eta_r(\pi_E)$ is estimated from $E$.

**Generative Adversarial Imitation Learning (GAIL)** Ho & Ermon (2016) draws a connection between IRL and Generative Adversarial Nets (GANs) as shown in Eq.1. A discriminator $D : \mathbb{S} \times \mathbb{A} \to [0,1]$ is trained by minimizing Eq.1 so that $D$ can accurately identify any $(s,a)$ generated by the agent. Meanwhile, an agent policy $\pi$ is trained as a generator by using a reward function induced from $D$ to maximize Eq.1 so that $D$ cannot discriminate $\tau \sim \pi$ from $\tau_E$. In adversarial inverse reinforcement learning (AIRL) Fu et al. (2018), it is further proposed that by representing $D(s,a) := \frac{\pi(a|s)}{\exp(r(s,a)) + \pi(a|s)}$ with $r$, when Eq.1 is at optimality, $r^* \equiv \log \pi_E \equiv \mathcal{A}_{\pi_E}$. By training

$\pi$ with $r^*$ until optimality, $\pi$ will behave just like $\pi_E$.

$$\mathbb{E}_{(s,a)\sim\pi}\left[\log(1 - D(s,a))\right] + \mathbb{E}_{(s,a)\sim\pi_E}\left[\log D(s,a)\right] \tag{1}$$

# 4  PROTAGONIST ANTAGONIST GUIDED ADVERSARIAL REWARD (PAGAR)

This section introduces our semi-supervised reward design paradigm, PAGAR. We first present the basic concepts of PAGAR and theoretically analyze the effect of applying PAGAR to RL. Then, we show how PAGAR can be incorporated with IRL-based IL to mitigate reward misalignment.

We call the policy to be trained for the underlying task the *protagonist policy* $\pi_P$. Recall that $U_r(\pi)$ mentioned in Definition 1 is used to measure the performance of a $\pi$ under an $r$. We consider using the expected return to measure $U_r(\pi)$, i.e., $U_r(\pi) := \eta_r(\pi)$. For any reward function $r \in R$, we define *Protagonist Antagonist Induced Regret* as in Eq.2 where $\pi_A$ is referred to as an *antagonist policy*. The intuition is that $\pi_P$ does not incur high regret under $r$ if its performance is close to that of the optimal policy under $r$. We define our semi-supervised reward design paradigm in Definition 2.

$$Regret(\pi_P, r) := \left\{\max_{\pi_A \in \Pi} U_r(\pi_A)\right\} - U_r(\pi_P) \tag{2}$$

**Definition 2** (Protagonist Antagonist Guided Adversarial Reward (PAGAR)). Given a candidate reward function set $R$ and a protagonist policy $\pi_P$, PAGAR searches for a reward function $r$ within $R$ to maximize the *Protagonist Antagonist Induced Regret*, i.e., $\max_{r \in R} Regret(r, \pi_P)$.

## 4.1  RL WITH PAGAR

We next show that, under certain conditions, by training the protagonist policy $\pi_P$ with the reward functions induced from PAGAR as shown by the objective function $MinimaxRegret$ in Eq.3, $\pi_P$ can guarantee to avoid failing and even accomplish the underlying task even when $R$ contains reward functions that are misaligned with the task. We denote any task-aligned reward function as $r_{al} \in R$ and any misaligned reward function as $r_{mis} \in R$. Then we use the $S_r$ and $F_r$ intervals defined in Definition 1 to present these conditions in Theorem 1.

$$MinimaxRegret(R) := \arg\min_{\pi_P \in \Pi}\max_{r \in R}\left\{\max_{\pi_A \in \Pi} U_r(\pi_A)\right\} - U_r(\pi_P) \tag{3}$$

**Theorem 1** (Task-Failure Avoidance ). *If the following conditions (1) (2) hold for $R$, then the optimal protagonist policy $\pi_P := MinimaxRegret(R)$ satisfies that $\forall r_{al} \in R, U_{r_{al}}(\pi_P) \notin F_{r_{al}}$.*

*(1) There exists $r_{al} \in R$, and $\max_{r_{al} \in R}\{\sup F_{r_{al}} - \inf F_{r_{al}}\} < \min_{r_{al} \in R}\{\inf S_{r_{al}} - \sup F_{r_{al}}\} \wedge$
    $\max_{r_{al} \in R}\{\sup S_{r_{al}} - \inf S_{r_{al}}\} < \min_{r_{al} \in R}\{\inf S_{r_{al}} - \sup F_{r_{al}}\}$;*

*(2) There exists a policy $\pi^*$ such that $\forall r_{al} \in R, U_{r_{al}}(\pi^*) \in S_{r_{al}}$, and $\forall r_{mis} \in R$,
    $\max_{\pi \in \Pi} U_{r_{mis}}(\pi) - U_{r_{mis}}(\pi^*) < \min_{r_{al} \in R}\{\inf S_{r_{al}} - \sup F_{r_{al}}\}$.*

In Theorem 1, condition (1) states that for each $r_{al} \in R$, the ranges of the utilities of successful and failing policies are distributed in small ranges. Condition (2) states that there exists a $\pi^*$ that not only performs well under all $r_{al}$'s (thus succeeding in the task), but also achieves high performance under all $r_{mis}$'s. The proof can be found in Appendix A.3. Furthermore, under stricter conditions, $MinimaxRegret(R)$ can guarantee inducing a policy that succeeds in the underlying task.

**Corollary 1** (Task-Success Guarantee). *Assume that Condition (1) in Theorem 1 is satisfied. If there exists a policy $\pi^*$ such that $\forall r \in R, \max_{\pi \in \Pi} U_r(\pi) - U_r(\pi^*) < \min_{r_{al} \in R}\{\sup S_{r_{al}} - \inf S_{r_{al}}\}$, then the optimal protagonist policy $\pi_P := MinimaxRegret(R)$ satisfies that $\forall r_{al} \in R, U_{r_{al}}(\pi) \in S_{r_{al}}$.*

The additional condition in Corollary 1 strengthens the requirement on the existence of a policy $\pi^*$ performing well under all reward functions in $R$. However, the theories raise the question of how to design an $R$ to meet those conditions without knowing which reward function aligns with the task. $R$ should not be chosen arbitrarily. For instance, if $R$ contains opposite reward functions $r$ and $-r$, no policy may perform well under both $r$ and $-r$. We next show that PAGAR can be effectively used in IL by associating $R$ with demonstrations that accomplish the task.

## 4.2 IL WITH PAGAR

Given a set $E$ of expert demonstrations, we let $R_{E,\delta} = \{r | U_r(E) - \max_{\pi \in \Pi} U_r(\pi) \geq \delta\}$ be the set of reward functions under which the performance margin between the expert demonstrations and the optimal policies is no less than $\delta$. Let $L_r$ be the Lipschitz constant of $r(\tau)$, $W_E$ be the smallest Wasserstein 1-distance $W_1(\pi, E)$ between $\tau \sim \pi$ of any $\pi$ and $\tau \sim E$, i.e., $W_E \triangleq \min_{\pi \in \Pi} W_1(\pi, E)$.

Then, we have the following.

**Theorem 2** (Task-Failure Avoidance). *If the following conditions (1) (2) hold for $R_{E,\delta}$, then the optimal protagonist policy $\pi_P := MinimaxRegret(R_{E,\delta})$ satisfies that $\forall r_{al} \in R_{E,\delta}$, $U_{r_{al}}(\pi) \notin F_{r_{al}}$.*

*(1) The condition (1) in Theorem 1 holds*

*(2) $\forall r_{al} \in R_{E,\delta}$, $L_{r_{al}} \cdot W_E - \delta \leq \sup S_{r_{al}} - \inf S_{r_{al}}$ and $\forall r_{mis} \in R_{E,\delta}$, $L_{r_{mis}} \cdot W_E - \delta < \min_{r_{al} \in R_{E,\delta}} \{\inf S_{r_{al}} - \sup F_{r_{al}}\}$.*

Theorem 2 delivers the same guarantee as that of Theorem 1 but differs from Theorem 1 in that condition (2) implies that there exists a policy $\pi^*$ such that the performance margin between $E$ and $\pi^*$ is small under all $r \in R_{E,\delta}$. The following corollary further describes a sufficient condition for $MinimaxRegret(R_{E,\delta})$ to find a policy that succeeds in the underlying task.

**Corollary 2** (Task-Success Guarantee). *Assume that the condition (1) in Theorem 1 holds for $R_{E,\delta}$. If for any $r \in R_{E,\delta}$, $L_r \cdot W_E - \delta \leq \min_{r_{al} \in R_{E,\delta}} \{\sup S_{r_{al}} - \inf S_{r_{al}}\}$, then the optimal protagonist policy $\pi_P = MinimaxRegret(R_{E,\delta})$ satisfies $\forall r_{al} \in R_{E,\delta}$, $U_{r_{al}}(\pi) \in S_{r_{al}}$.*

Corollary 2 suggests using a large $\delta$ to gain a better chance of satisfying the conditions in Corollary 2. Notably, increasing the $\delta$ in $R_{E,\delta}$ coincides with decreasing the IRL loss $J_{IRL}$ given in Section 3 if not considering the entropy regularization. Specifically, by reformulating the IRL loss as $J_{IRL}(\pi, r) := U_r(\pi) - U_r(E)$, $r \in R_{E,\delta}$ can be enforced by $J_{IRL}(r) + \delta \leq 0$. Hence, we propose IL with PAGAR: *use IRL to learn a $R_{E,\delta}$ for some target $\delta$, and learn a $\pi_P$ via $MinimaxRegret(R_{E,\delta})$.* In particular, for the maximum $\delta := \max_{r \in R} U_r(E) - \max_{\pi \in \Pi} U_r(\pi)$, we simplify $R_{E,\delta}$ as $R_E$. In this case, $R_E$ equals the optimal reward function solution set of IRL, and solving $MinimaxRegret(R_E)$ can circumvent the reward ambiguity issue in IRL. We use an example to illustrate this.

**Example 1.** In a two-state transition system where $s_0$ is the initial state, and $s_1$ is an absorbing state, an agent can choose action $a_0$ at state $s_0$ to stay at $s_0$ or choose $a_1$ to reach $s_1$. Any agent can start from $s_0$ and choose actions for 5 timesteps. The task is to learn a stochastic policy to reach $s_1$. Expert only demonstrates one trajectory $E = \{\tau_E = (s_0, a_1, s_1, s_1, s_1, s_1)\}$, i.e., choose $a_1$ at $s_0$ and then stay in $s_1$ for the rest 4 steps. The convex combinations of two basis functions $r_1$ and $r_2$ constitute the hypothesis set of reward functions $R = \{r | r(s, a) = \alpha \cdot r_1(s, a) + (1 - \alpha) \cdot r_2(s, a), \alpha \in [0, 1]\}$ where $r_1(s, a) \equiv 4$ constantly, and $r_2(s_0, a_0) = r_2(s_0, a_1) = 0, r_2(s_1, \cdot) = 5$. We prove in Appendix A.5 that when applying IRL to learn a reward function from $E$, any convex combination of $r_1$ and $r_2$ is an optimal solution, i.e., $R_E = R$. This ambiguity in choosing reward functions can cause reward misalignment because the reward function $r = r_1$ induced by $\alpha = 1$ violates the criterion of task-reward alignment in Definition 1: achieving optimality under $r_1$ does not guarantee task success since all policies are optimal under $r_1$. On the other hand, $MinimaxRegret(R)$ will produce a policy $\pi_P(a_1|s_0) = 1$ of which the worst-case regret is 0. And it is the desired solution.

Table 1 compares PAGAR-based IL and IRL-based IL from another perspective by deriving an equivalent objective function of $MinimaxRegret(R_E)$. The derivation is shown in Theorem 5 in Appendix A.2. In essence, this objective function searches for a policy $\pi$ with the highest score measured by an affine combination of policy performance $U_r(\pi)$ with $r$ drawn from two different reward function distributions in $R_E$. One distribution has singleton support on the reward function $r_\pi^*$ that maximizes the policy performance $U_r(\pi)$ among those who maximize the regret $Regret(\pi, r)$. The other one, $\mathcal{P}_\pi$, is a baseline distribution which guarantees that: (1) for policies that do not always perform worse than any other policy, the expected $U_r(\pi)$ values measured under $r \sim \mathcal{P}_\pi$ are all equal to some constant $c$ (minimum value for the equality to hold); (2) for any other policy $\pi'$,

| | |
|---|---|
| IRL-Based IL | $\arg\max_{\pi \in \Pi} U_r(\pi) \qquad s.t.\ r \in \arg\min_{r' \in R} \left\{ \max_{\pi' \in \Pi} U_{r'}(\pi) \right\} - U_{r'}(\pi_E)$ |
| PAGAR-Based IL | $\arg\max_{\pi \in \Pi} \left\{ \frac{Regret(\pi, r_\pi^*)}{c - U_{r_\pi^*}(\pi)} \cdot U_{r_\pi^*}(\pi) + \mathbb{E}_{r \sim \mathcal{P}_\pi(r)} \left[ \left( 1 - \frac{Regret(\pi, r)}{c - U_r(\pi)} \right) \cdot U_r(\pi) \right] \right\}$ |

Table 1: Comparing the policy optimization objectives of IRL-based IL and PAGAR-based IL. For PAGAR, $\pi$ is the protagonist policy; $\mathcal{P}_\pi(r)$ is a baseline distribution over $r \in R_E$ such that 1) $c$ is the smallest value for $c \equiv \mathbb{E}_{r \sim \mathcal{P}_\pi} [U_r(\pi)]$ to hold for all the policies that perform no worse than any other policy, and 2) $\max_{r \in R_E} U_r(\pi) \equiv \mathbb{E}_{r \sim \mathcal{P}_\pi} [U_r(\pi)]$ holds for any other policy; $r_\pi^*$ is a reward function $r$ associated with a policy $\pi$ such that $r_\pi^*$ maximizes $U_r(\pi)$ among all those $r$'s that maximizes $Regret(\pi, r)$. Details can be found in the Appendix A.1 and A.2.

the distribution concentrates on the reward function $r'$ under which the policy achieves the highest performance $U_{r'}(\pi')$. The existence of such $\mathcal{P}_\pi$ is proven in Appendix A.1 and A.2. Intuitively, the affine combination assigns different weights to the policy performances evaluated under those two distributions. If the policy $\pi$ performs worse under $r_\pi^*$ than under many other reward functions ($U{r_\pi^*}(\pi)$ falls below $c$), a higher weight will be allocated to using $r_\pi^*$ to train $\pi$. Conversely, if the policy $\pi$ performs better under $r_\pi^*$ than under many other reward functions ($c$ falls below $U{r_\pi^*}(\pi)$), a higher weight will be allocated to using reward functions drawn from $\mathcal{P}_\pi$ to train $\pi$. Furthermore, if there exists a policy that is optimal under all the optimal reward functions in $R_E$, i.e., IRL can reach Nash Equilibrium, $MinimaxRegret(R_E)$ will choose this policy as its solution. In other words, $MinimaxRegret(R_E)$ does not degrade the performance of IRL.

**Theorem 3.** *If* $\arg\min_{r \in R} \max_{\pi \in \Pi} J_{IRL}(R, \pi)$ *can reach Nash Equilibrium with an optimal reward function solution set* $R_E$ *and an optimal policy set* $\Pi_E$, *then* $\Pi_E$ *is equal to the set of solutions to* $MinimiaxRegret(R_E)$.

## 5  AN ON-AND-OFF-POLICY APPROACH TO PAGAR-BASED IL

In this section, we introduce a practical approach to solving $MinimaxRegret(R_{E,\delta})$ with given $\delta$. In a nutshell, this approach alternates between policy learning and reward learning. We first explain how we optimize $\pi_P, \pi_A$; then we derive from Eq.2 two reward improvement bounds for optimizing $r$. We then discuss how to incorporate IRL to enforce the constraint $r \in R_{E,\delta}$.

### 5.1  POLICY OPTIMIZATION WITH ON-AND-OFF POLICY SAMPLES

Following the entropy-regularized RL framework, we adopt $U_r(\pi) := \eta_r(\pi) + \mathcal{H}(\pi)$, which is equal to the RL objective $J_{RL}(\pi; r)$ as defined in Section 3. Given an intermediate learned reward function $r$, according to $MinimaxRegret$, the objective function for optimizing $\pi_P$ is $\max_{\pi_P} \eta_r(\pi_P) - \eta_r(\pi_A) + \mathcal{H}(\pi_P)$. According to Schulman et al. (2015), $\eta_r(\pi_P) - \eta_r(\pi_A) \geq \sum_{s \in \mathbb{S}} \rho_{\pi_A}(s) \sum_{a \in \mathbb{A}} \pi_P(a|s) \hat{\mathcal{A}}_{\pi_A}(s, a) - C \cdot \max_s D_{TV}(\pi_A(\cdot|s), \pi_P(\cdot|s))^2$ where $\rho_{\pi_A}(s) = \sum_{t=0}^T \gamma^t Prob(s^{(t)} = s|\pi_A)$ is the discounted visitation frequency of $\pi_A$, $\hat{\mathcal{A}}_{\pi_A}(s, a)$ is the advantage function when the entropy regularizer $\mathcal{H}$ is not considered, and $C$ is some constant. This inequality allows us to maximize the $\eta_r(\pi_P) - \eta_r(\pi_A)$ part by only using the trajectories of $\pi_A$ (off-policy): following the theories in Schulman et al. (2015) and Schulman et al. (2017), we derive from the r.h.s of the inequality a PPO-style objective function $J_{PPO}(\pi_P; \pi_A, r) := \mathbb{E}_{s \sim \pi_A}[\min(\frac{\pi_P(a|s)}{\pi_A(a|s)} \hat{\mathcal{A}}_{\pi_A}(s, a), clip(\frac{\pi_P(a|s)}{\pi_A(a|s)}, 1 - \sigma, 1 + \sigma) \hat{\mathcal{A}}_{\pi_A}(s, a)]$ where $\sigma$ is a clipping threshold. In the meantime, since $\max_{\pi_P} \eta(\pi_P) - \eta_r(\pi_A) + \mathcal{H}(\pi_P) \equiv \max_{\pi_P} \eta(\pi_P) + \mathcal{H}(\pi_P)$, we can directly optimize $\pi_P$ by maximizing a standard RL objective $J_{RL}(\pi_P; r) = \eta_r(\pi_P) + \mathcal{H}(\pi_P)$ with the trajectories of $\pi_P$ itself (on-policy). Regarding the optimization of $\pi_A$, since $\pi_A$ is intended to be a proxy of the optimal policy under $r$, we train $\pi_A$ to maximize $J_{RL}(\pi_A; r)$. The loss functions for optimizing $\pi_A$ is denoted as $J_{RL}(\pi_A; r)$ and that of $\pi_P$ is $J_{PPO}(\pi_P; \pi_A, r) + J_{RL}(\pi_P; r)$.

**Algorithm 1** An On-and-Off-Policy Algorithm for Imitation Learning with PAGAR

**Input**: Expert demonstration $E$, margin upper-bound $\delta$, initial protagonist policy $\pi_P$, antagonist policy $\pi_A$, reward function $r$, Lagrangian parameter $\lambda \geq 0$, maximum iteration number $N$.

**Output**: $\pi_P$

1: **for** iteration $i = 0, 1, \ldots, N$ **do**
2:     Sample trajectory sets $\mathbb{D}_A \sim \pi_A$ and $\mathbb{D}_P \sim \pi_P$
3:     **Optimize** $\pi_A$: estimate $J_{RL}(\pi_A; r)$ with $\mathbb{D}_A$; maximize $J_{RL}(\pi_A; r)$
4:     **Optimize** $\pi_P$: estimate $J_{RL}(\pi_P; r)$ with $\mathbb{D}_P$; estimate $J_{PPO}(\pi_P; \pi_A, r)$ with $\mathbb{D}_A$; maximize $J_{RL}(\pi_P; r) + J_{PPO}(\pi_P; \pi_A, r)$
5:     **Optimize** $r$: estimate $J_{PAGAR}(r; \pi_P, \pi_A)$ with $\mathbb{D}_P$ and $\mathbb{D}_A$; estimate $J_{IRL}(r)$ with $\mathbb{D}_A$ and $E$; minimize $J_{PAGAR}(r; \pi_P, \pi_A) + \lambda \cdot (J_{IRL}(r) + \delta)$; then update $\lambda$ based on $J_{IRL}(r) + \delta$
6: **end for**
7: **return** $\pi_P$

## 5.2 REGRET MAXMIZATION WITH ON-AND-OFF POLICY SAMPLES

Given the intermediate learned protagonist and antagonist policy $\pi_P$ and $\pi_A$, according to $MinimaxRegret$, we need to optimize $r$ to maximize $Regret(r, \pi_P) = \max_{\pi \in \Pi} \eta_r(\pi) - \eta_r(\pi_P)$. In practice, we solve $\arg\max_r \eta_r(\pi_A) - \eta_r(\pi_P)$ since $\pi_A$ is the proxy of the optimal policy under $r$. We extract the two reward improvement bounds in Theorem 4 to help solve this objective function.

**Theorem 4.** *Suppose policy $\pi_2 \in \Pi$ is the optimal solution for $J_{RL}(\pi; r)$. Then for any policy $\pi_1 \in \Pi$, the inequalities Eq.4 and 5 hold where $\alpha = \max_s D_{TV}(\pi_1(\cdot|s), \pi_2(\cdot|s))$, $\epsilon = \max_{s,a} |\mathcal{A}_{\pi_2}(s, a)|$, and $\Delta\mathcal{A}(s) = \mathbb{E}_{a \sim \pi_1}[\mathcal{A}_{\pi_2}(s, a)] - \mathbb{E}_{a \sim \pi_2}[\mathcal{A}_{\pi_2}(s, a)]$.*

$$\left| \eta_r(\pi_1) - \eta_r(\pi_2) - \sum_{t=0}^{\infty} \gamma^t \mathbb{E}_{s^{(t)} \sim \pi_1}\left[\Delta\mathcal{A}(s^{(t)})\right] \right| \leq \frac{2\alpha\gamma\epsilon}{(1-\gamma)^2} \tag{4}$$

$$\left| \eta_r(\pi_1) - \eta_r(\pi_2) - \sum_{t=0}^{\infty} \gamma^t \mathbb{E}_{s^{(t)} \sim \pi_2}\left[\Delta\mathcal{A}(s^{(t)})\right] \right| \leq \frac{2\alpha\gamma(2\alpha + 1)\epsilon}{(1-\gamma)^2} \tag{5}$$

By letting $\pi_P$ be $\pi_1$ and $\pi_A$ be $\pi_2$, Theorem 4 enables us to bound $\eta_r(\pi_A) - \eta_r(\pi_P)$ by using either only the samples of $\pi_A$ or only those of $\pi_P$. Following Fu et al. (2018), we let $r$ be a proxy of $\mathcal{A}_{\pi_2}$ in Eq.4 and 5. Then we derive two loss functions $J_{R,1}(r; \pi_P, \pi_A)$ and $J_{R,2}(r; \pi_P, \pi_A)$ for $r$ as shown in Eq.6 and 7 where $\xi_1(s, a) = \frac{\pi_P(a|s)}{\pi_A(a|s)}$ and $\xi_2(s, a) = \frac{\pi_A(a|s)}{\pi_P(a|s)}$ are importance sampling ratios, and $C_1$ and $C_2$ are constants proportional to the estimated maximum KL divergence between $\pi_A$ and $\pi_P$ (to bound $\alpha$ Schulman et al. (2015)). The objective function for $r$ is then $J_{PAGAR} := J_{R,1} + J_{R,2}$.

$$J_{R,1}(r; \pi_P, \pi_A) := \mathbb{E}_{\tau \sim \pi_A}\left[\sum_{t=0}^{\infty} \gamma^t \left(\xi_1(s^{(t)}, a^{(t)}) - 1\right) \cdot r(s^{(t)}, a^{(t)})\right] + C_1 \cdot \max_{(s,a) \sim \pi_A} |r(s, a)| \tag{6}$$

$$J_{R,2}(r; \pi_P, \pi_A) := \mathbb{E}_{\tau \sim \pi_P}\left[\sum_{t=0}^{\infty} \gamma^t \left(1 - \xi_2(s^{(t)}, a^{(t)})\right) \cdot r(s^{(t)}, a^{(t)})\right] + C_2 \cdot \max_{(s,a) \sim \pi_P} |r(s, a)| \tag{7}$$

## 5.3 ALGORITHM FOR SOLVING PAGAR-BASED IL

In addition to $J_{PAGAR}(r; \pi_P, \pi_A)$, we incorporate IRL to enforce the constraint $r \in R_{E,\delta}$ by adding a penalty term $\lambda \cdot (\delta + J_{IRL})$ where $\lambda$ is a Lagrangian parameter. We then reformulate the objective function for optimizing $r$ as $\min_{r \in R} J_{PAGAR}(r; \pi_P, \pi_A) + \lambda \cdot (\delta + J_{IRL}(r))$. Particularly, for some tasks, we solve $\min_r J_{PAGAR} - \lambda \cdot J_{IRL}$ with a large constant $\lambda$ to equivalently implement the maximal $\delta$ as mentioned in Section 4.2. Algorithm 1 describes our approach for solving PAGAR-based IL. The algorithm iteratively trains the policies and the reward function alternately. It first trains $\pi_A$ in line 3. Then, it employs the on-and-off policy approach to train $\pi_P$ in line 4, including utilizing the PPO-style objective $J_{PPO}$. In line 5, while $J_{PAGAR}$ is estimated based on both $\mathbb{D}_A$ and $\mathbb{D}_P$, the IRL objective is only based on $\mathbb{D}_A$. Appendix B.3 details how we update $\lambda$ based on $J_{IRL}$ and incorporate different IRL algorithms.

# 6 EXPERIMENTS

The goal of our experiments is to assess whether using PAGAR-based IL can efficiently circumvent reward misalignment in different IL/IRL benchmarks by comparing with representative baselines. We present the main results below and provide details and additional results in Appendix C.

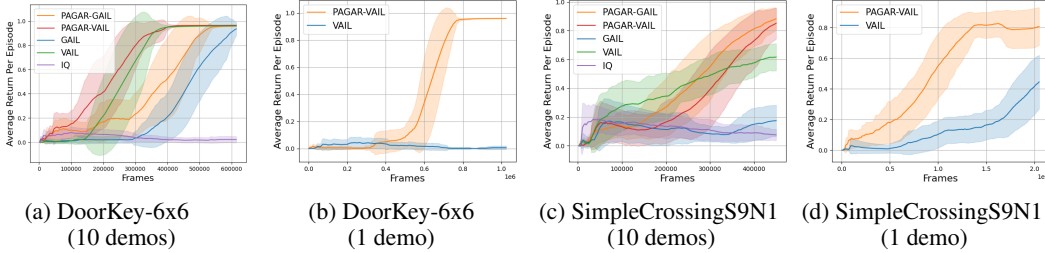

| (a) DoorKey-6x6 (10 demos) | (b) DoorKey-6x6 (1 demo) | (c) SimpleCrossingS9N1 (10 demos) | (d) SimpleCrossingS9N1 (1 demo) |

Figure 2: Comparing Algorithm 1 with baselines in two partial observable navigation tasks. The suffix after each 'PAGAR-' indicates which IRL technique is used in Algorithm 1. The $y$ axis indicates the average return per episode. The $x$ axis indicates the number of timesteps.

## 6.1 PARTIALLY OBSERVABLE NAVIGATION TASKS

We first consider a maze navigation environment where the task objective can be straightforwardly categorized as either success or failure. Our benchmarks include two discrete domain tasks from the Mini-Grid environments Chevalier-Boisvert et al. (2023): *DoorKey-6x6-v0*, and *SimpleCrossingS9N1-v0*. Due to partial observability and the implicit hierarchical nature of the task, these environments are considered challenging for RL and IL, and have been extensively used for benchmarking curriculum RL and exploration-driven RL. In *DoorKey-6x6-v0* the task is to pick up a key, unlock a door, and reach a target position; in *SimpleCrossingS9N1*, the task is to pass an opening on a wall and reach a target position. The placements of the objects, obstacles, and doors are randomized in each instance of an environment. The agent can only observe a small, unblocked area in front of it. At each timestep, the agent can choose one out of 7 actions, such as moving to the next cell or picking up an object. By default, the reward is always zero unless the agent reaches the target. We compare our approach with two competitive baselines: GAIL Ho & Ermon (2016) and VAIL Peng et al. (2019). GAIL has been introduced in Section 3. VAIL is based on GAIL but additionally optimizes a variational discriminator bottleneck (VDB) objective. Our approach uses the IRL techniques behind those two baseline algorithms, resulting in two versions of Algorithm 1, denoted as PAGAR-GAIL and PAGAR-VAIL, respectively. More specifically, if the baseline optimizes a $J_{IRL}$ objective, we use the same $J_{IRL}$ objective in Algorithm 1. Also, we represent the reward function $r$ with the discriminator $D$ as mentioned in Section 3. More details can be found in Appendix C.1. PPO Schulman et al. (2017) is used for policy training in GAIL, VAIL, and ours. Additionally, we compare our algorithm with a state-of-the-art (SOTA) IL algorithm, IQ-Learn Garg et al. (2021), which, however, is not compatible with our algorithm because it does not explicitly optimize a reward function. We use a replay buffer of size $2048$ in our algorithm and all the baselines. The policy and the reward functions are all approximated using convolutional networks. By learning from 10 expert-demonstrated trajectories with high returns, PAGAR-based IL produces high-performance policies with high sample efficiencies as shown in Figure 2(a) and (c). Furthermore, we compare PAGAR-VAIL with VAIL by reducing the number of demonstrations to one. As shown in Figure 2(b) and (d), PAGAR-VAIL produces high-performance policies with significantly higher sample efficiencies.

**Zero-Shot IL in Transfer Environments.** In this experiment, we show that PAGAR can enable the agent to infer and accomplish the objective of a task in environments that are substantially different from the expert demonstration environment. As shown in Figure 3(a), by using the 10 expert demonstrations in *SimpleCrossingS9N1-v0*, we apply Algorithm 1 and the baselines, GAIL, VAIL, and IQ-learn to learn policies in *SimpleCrossingS9N2-v0, SimpleCrossingS9N3-v0* and *FourRooms-v0*. The results in Figure 3(b)-(d) show that PAGAR-based IL outperforms the baselines in these challenging zero-shot settings.

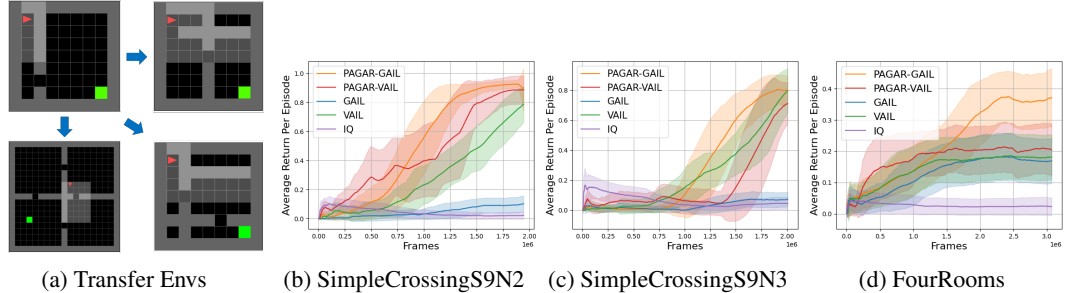

| (a) Transfer Envs | (b) SimpleCrossingS9N2 | (c) SimpleCrossingS9N3 | (d) FourRooms |

Figure 3: Comparing Algorithm 1 with baselines in transfer environments.

**Influence of Reward Hypothesis Space**. We study whether choosing a different reward function hypothesis set can influence the performance of Algorithm 1. Specifically, we compare using a $Sigmoid$ function with a Categorical distribution in the output layer of the discriminator networks in GAIL and PAGAR-GAIL. When using the $Sigmoid$ function, the outputs of $D$ are not normalized, i.e., $\sum_{a \in \mathbb{A}} D(s, a) \neq 1$. When using a Categorical distribution, the outputs in a state sum to one for all the actions, i.e., $\sum_{a \in \mathbb{A}} D(s, a) = 1$. As a result, the sizes of the reward function sets are different in the two cases. We test GAIL and PAGAR-GAIL in *DoorKey-6x6-v0* environment. As shown in Figure 4, different reward function sets result in different training efficiency. However, PAGAR-GAIL outperforms GAIL in both cases by using fewer samples to attain high performance.

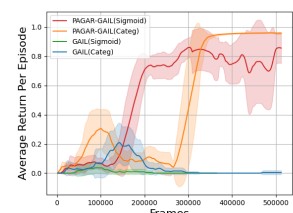

Figure 4: DoorKey-6x6

### 6.2 CONTINUOUS CONTROL TASKS WITH NON-BINARY OUTCOMES

We test PAGAR-based IRL in multiple Mujuco tasks: *Walker2d-v2, HalfCheetah-v2, Hopper-v2, InvertedPendulum-v2, and Swimmer-v2* where the task objectives do not have binary outcomes.

In Figure 5, we show the results on two tasks (the other results are included in Appendix C.3). The results show that PAGAR-based IL takes fewer iterations to achieve the same performance as the baselines. In particular, in the *HalfCheetah-v2* task, Algorithm 1 achieves the same level of performance compared with GAIL and VAIL by using only half the number of iterations. We note that IQ-learn could perform better if a much larger replay buffer were used since it uses the off-policy RL algorithm SAC Haarnoja et al. (2018).

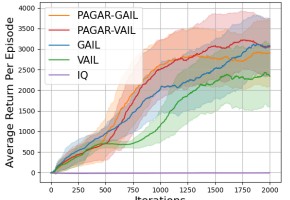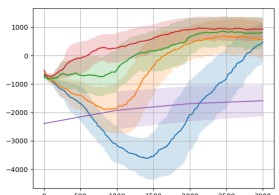

Figure 5: (Left: Walker2d-v2. Right: HalfCheeta-v2) The $y$ axis indicates the average return per episode. One exception is IQ-Learn, which updates the policy at every timestep, making its actual number of iterations 2048 times larger than in the figures.

## 7 CONCLUSION

We propose PAGAR, a semi-supervised reward design paradigm that generates adversarial reward functions under the guidance of a protagonist policy and an antagonist policy. PAGAR-based IL can overcome the reward misalignment problem of IRL-based IL by training a policy that performs well under multiple adversarially selected reward functions. We present an on-and-off policy approach to PAGAR-based IL by using policy and reward improvement bounds to maximize the utilization of policy samples. Experimental results demonstrate that our algorithm can mitigate reward misalignment in challenging environments. Our future work will focus on reducing the computational overhead in policy training and accommodating the PAGAR paradigm in other IL settings.

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

APPENDIX

In the appendix, we provide additional details of the theories and the experiments. The contents of this appendix are as follows.

- In Appendix A, we discuss some details of PAGAR and $MinimaxRegret$ that were omitted in Section 4. We briefly introduce some necessary preliminaries in Appendix A.1. Then we derive a Theorem 5 to support **Table 1** in Appendix A.2. The proves for **Theorem** 1 and 2, **Corollary** 1 and 3, **Theorem** 3 are in Appendix A.4. The details of **Example** 1 are in Appendix A.5.
- In Appendix B, we provide some details of Imitation Learning with PAGAR that were omitted in Section 5. We prove **Theorem**4 in Appendix B.1. Then we derive the objective functions in Appendix B.2. Some details of Algorithm 1 will be explained in Appendix B.3
- In Appendix C, we provide some experimental details and additional results.

## A   REWARD DESIGN WITH PAGAR

This paper does not aim to resolve the ambiguity problem in IRL but provides a way to circumvent it so that reward ambiguity does not lead to reward misalignment in IRL-based IL. PAGAR, the semi-supervised reward design paradigm proposed in this paper, tackles this problem from the perspective of semi-supervised reward design. But the nature of PAGAR is distinct from IRL and IL: assume that a set of reward functions is available for some underlying task, where some of those reward functions align with the task while others are misaligned, PAGAR provides a solution for selecting reward functions to train a policy that successfully performs the task, without knowing which reward function aligns with the task. Our research demonstrates that policy training with PAGAR is equivalent to learning a policy to maximize an affine combination of utilities measured under a distribution of the reward functions in the reward function set. With this understanding of PAGAR, we integrate it with IL to illustrate its advantages.

### A.1   SEMI-SUPERVISED REWARD DESIGN

Designing a reward function can be thought as deciding an ordering of policies. We adopt a concept, called *total domination*, from unsupervised environment design Dennis et al. (2020), and re-interpret this concept in the context of reward design. In this paper, we suppose that a function $U_r(\pi)$ is given to measure the performance of a policy. While the measurement of policy performance can vary depending on the free variable $r$, *total dominance* can be viewed as an invariance regardless of such dependency.

**Definition 3** (Total Domination). A policy, $\pi_1$, is totally dominated by some policy $\pi_2$ w.r.t a reward function set $R$, if for every pair of reward functions $r_1, r_2 \in R$, $U_{r_1}(\pi_1) < U_{r_2}(\pi_2)$.

If $\pi_1$ totally dominate $\pi_2$ w.r.t $R$, $\pi_2$ can be regarded as being unconditionally better than $\pi_1$. In other words, the two sets $\{U_r(\pi_1)|r \in R\}$ and $\{U_r(\pi_2)|r \in R\}$ are disjoint, such that $\sup\{U_r(\pi_1)|r \in R\} < \inf\{U_r(\pi_2)|r \in R\}$. Conversely, if a policy $\pi$ is not totally dominated by any other policy, it indicates that for any other policy, say $\pi_2$, $\sup\{U_r(\pi_1)|r \in R\} \geq \inf\{U_r(\pi_2)|r \in R\}$.

**Definition 4.** A reward function set $R$ aligns with an ordering $\prec_R$ among policies such that $\pi_1 \prec_R \pi_2$ if and only if $\pi_1$ is totally dominated by $\pi_2$ w.r.t. $R$.

Especially, designing a reward function $r$ is to establish an ordering $\prec_{\{r\}}$ among policies. Total domination can be extended to policy-conditioned reward design, where the reward function $r$ is selected by following a decision rule $\mathcal{R}(\pi)$ such that $\sum_{r \in R} \mathcal{R}(\pi)(r) = 1$. We let $\mathcal{U}_\mathcal{R}(\pi) = \sum_{r \in R_E} \mathcal{R}(\pi)(r) \cdot U_r(\pi)$ be an affine combination of $U_r(\pi)$'s with its coefficients specified by $\mathcal{R}(\pi)$.

**Definition 5.** A policy conditioned decision rule $\mathcal{R}$ is said to prefer a policy $\pi_1$ to another policy $\pi_2$, which is notated as $\pi_1 \prec^\mathcal{R} \pi_2$, if and only if $\mathcal{U}_\mathcal{R}(\pi_1) < \mathcal{U}_\mathcal{R}(\pi_2)$.

Making a decision rule for selecting reward functions from a reward function set to respect the total dominance w.r.t this reward function set is an unsupervised learning problem, where no additional

external supervision is provided. If considering expert demonstrations as a form of supervision and using it to constrain the set $R_E$ of reward function via IRL, the reward design becomes semi-supervised.

## A.2  SOLUTION TO THE MINIMAXREGRET

In Table 1, we mentioned that solving $MinimaxRegret(R_E)$ is equivalent to finding an optimal policy $\pi^*$ to maximize a $\mathcal{U}_{\mathcal{R}_E}(\pi)$ under a decision rule $\mathcal{R}_E$. Without loss of generality, we use $R$ instead of $R_E$ in our subsequent analysis, because solving $MinimaxRegret(R)$ does not depend on whether there are constraints for $R$. In order to show such an equivalence, we follow the same routine as in Dennis et al. (2020), and start by introducing the concept of *weakly total domination*.

**Definition 6** (Weakly Total Domination). A policy $\pi_1$ is *weakly totally dominated* w.r.t a reward function set $R$ by some policy $\pi_2$ if and only if for any pair of reward function $r_1, r_2 \in R$, $U_{r_1}(\pi_1) \leq U_{r_2}(\pi_2)$.

Note that a policy $\pi$ being totally dominated by any other policy is a sufficient but not necessary condition for $\pi$ being weakly totally dominated by some other policy. A policy $\pi_1$ being weakly totally dominated by a policy $\pi_2$ implies that $\sup\{U_r(\pi_1)|r \in R\} \leq \inf\{U_r(\pi_2)|r \in R\}$. We assume that there does not exist a policy $\pi$ that weakly totally dominates itself, which could happen if and only if $U_r(\pi)$ is a constant. We formalize this assumption as the following.

**Assumption 1.** For the given reward set $R$ and policy set $\Pi$, there does not exist a policy $\pi$ such that for any two reward functions $r_1, r_2 \in R$, $U_{r_1}(\pi) = U_{r_2}(\pi)$.

This assumption makes weak total domination a non-reflexive relation. It is obvious that weak total domination is transitive and asymmetric. Now we show that successive weak total domination will lead to total domination.

**Lemma 1.** *for any three policies $\pi_1, \pi_2, \pi_3 \in \Pi$, if $\pi_1$ is weakly totally dominated by $\pi_2$, $\pi_2$ is weakly totally dominated by $\pi_3$, then $\pi_3$ totally dominates $\pi_1$.*

*Proof.* According to the definition of weak total domination, $\max_{r \in R} U_r(\pi_1) \leq \min_{r \in R} U_r(\pi_2)$ and $\max_{r \in R} U_r(\pi_2) \leq \min_{r \in R} U_r(\pi_3)$. If $\pi_1$ is weakly totally dominated but not totally dominated by $\pi_3$, then $\max_{r \in R} U_r(\pi_1) = \min_{r \in R} U_r(\pi_3)$ must be true. However, it implies $\min_{r \in R} U_r(\pi_2) = \max_{r \in R} U_r(\pi_2)$, which violates Assumption 1. We finish the proof. $\square$

**Lemma 2.** *For the set $\Pi_{\neg wtd} \subseteq \Pi$ of policies that are not weakly totally dominated by any other policy in the whole set of policies w.r.t a reward function set $R$, there exists a range $\mathbb{U} \subseteq \mathbb{R}$ such that for any policy $\pi \in \Pi_{\neg wtd}$, $\mathbb{U} \subseteq [\min_{r \in R} U_r(\pi), \max_{r \in R} U_r(\pi)]$.*

*Proof.* For any two policies $\pi_1, \pi_2 \in \Pi_{\neg wtd}$, it cannot be true that $\max_{r \in R} U_r(\pi_1) = \min_{r \in R} U_r(\pi_2)$ nor $\min_{r \in R} U_r(\pi_1) = \max_{r \in R} U_r(\pi_2)$, because otherwise one of the policies weakly totally dominates the other. Without loss of generalization, we assume that $\max_{r \in R} U_r(\pi_1) > \min_{r \in R} U_r(\pi_2)$. In this case, $\max_{r \in R} U_r(\pi_2) > \min_{r \in R} U_r(\pi_1)$ must also be true, otherwise $\pi_1$ weakly totally dominates $\pi_2$. Inductively, $\min_{\pi \in \Pi_{\neg wtd}} \max_{r \in R} U_r(\pi) > \max_{\pi \in \Pi_{\neg wtd}} \min_{r \in R} U_r(\pi)$. Letting $ub = \min_{\pi \in \Pi_{\neg wtd}} \max_{r \in R} U_r(\pi)$ and $lb = \max_{\pi \in \Pi_{\neg wtd}} \min_{r \in R} U_r(\pi)$, any $\mathbb{U} \subseteq [lb, ub]$ shall support the assertion. We finish the proof. $\square$

**Lemma 3.** *For a reward function set $R$, if a policy $\pi \in \Pi$ is weakly totally dominated by some other policy in $\Pi$ and there exists a subset $\Pi_{\neg wtd} \subseteq \Pi$ of policies that are not weakly totally dominated by any other policy in $\pi$, then $\max_{r \in R} U_r(\pi) < \min_{\pi' \in \Pi_{\neg wtd}} \max_{r \in R} U_r(\pi')$*

*Proof.* If $\pi_1$ is weakly totally dominated by a policy $\pi_2 \in \Pi$, then $\min_{r \in R} U_r(\pi_2) = \max_{r \in R} U_r(\pi)$. If $\max_{r \in R} U_r(\pi) \geq \min_{\pi' \in \Pi_{\neg wtd}} \max_{r \in R} U_r(\pi')$, then $\min_{r \in R} U_r(\pi_2) \geq \min_{\pi' \in \Pi_{\neg wtd}} \max_{r \in R} U_r(\pi')$, making at

least one of the policies in $\Pi_{\neg wtd}$ being weakly totally dominated by $\pi_2$. Hence, $\max\limits_{r \in R} U_r(\pi) <$
$\min\limits_{\pi' \in \Pi_{\neg wtd}} \max\limits_{r \in R} U_r(\pi')$ must be true. $\qquad\qquad\qquad\qquad\qquad\qquad\qquad\qquad\qquad\qquad\qquad$ $\square$

Given a policy $\pi$ and a reward function $r$, the regret is represented as Eq.8

$$Regret(\pi, r) \quad := \quad \max_{\pi'} U_r(\pi') - U_r(\pi) \tag{8}$$

Then we represent the $MinimaxRegret(R)$ problem in Eq.9.

$$MinimaxRegret(R) \quad := \quad \arg\min_{\pi \in \Pi} \left\{ \max_{r \in R} Regret(\pi, r) \right\} \tag{9}$$

We denote as $r_\pi^* \in R$ the reward function that maximizes $U_r(\pi)$ among all the $r$'s that achieve the maximization in Eq.9. Formally,

$$r_\pi^* \quad \in \quad \arg\max_{r \in R} U_r(\pi) \qquad s.t.\ r \in \arg\max_{r' \in R} Regret(\pi, r') \tag{10}$$

Then $MinimaxRegret$ can be defined as minimizing the worst-case regret as in Eq.9. Next, we want to show that for some decision rule $\mathcal{R}$, the set of optimal policies which maximizes $\mathcal{U}_\mathcal{R}$ are the solutions to $MinimaxRegret(R)$. Formally,

$$MinimaxRegret(R) = \arg\max_{\pi \in \Pi} \mathcal{U}_\mathcal{R}(\pi) \tag{11}$$

We design $\mathcal{R}$ by letting $\mathcal{R}(\pi) := \overline{\mathcal{R}}(\pi) \cdot \delta_{r_\pi^*} + (1 - \overline{\mathcal{R}}(\pi)) \cdot \underline{\mathcal{R}}(\pi)$ where $\underline{\mathcal{R}} : \Pi \to \Delta(R)$ is a policy conditioned distribution over reward functions, $\delta_{r_\pi^*}$ be a delta distribution centered at $r_\pi^*$, and $\overline{\mathcal{R}}(\pi)$ is a coefficient. We show how to design $\underline{\mathcal{R}}$ by using the following lemma.

**Lemma 4.** *Given that the reward function set is $R$, there exists a decision rule $\underline{\mathcal{R}} : \Pi \to \Delta(R)$ which guarantees that: 1) for any policy $\pi$ that is not weakly totally dominated by any other policy in $\Pi$, i.e., $\pi \in \Pi_{\neg wtd} \subseteq \Pi$, $\mathcal{U}_{\underline{\mathcal{R}}}(\pi) \equiv c$ where $c = \max\limits_{\pi' \in \Pi_{\neg wtd}} \min\limits_{r \in R} U_r(\pi')$; 2) for any $\pi$ that is weakly totally dominated by some policy but not totally dominated by any policy, $\mathcal{U}_{\underline{\mathcal{R}}}(\pi) = \max\limits_{r \in R} U_r(\pi)$; 3) if $\pi$ is totally dominated by some other policy, $\overline{\mathcal{R}}(\pi)$ is a uniform distribution.*

*Proof.* Since the description of $\underline{\mathcal{R}}$ for the policies in condition 2) and 3) are self-explanatory, we omit the discussion on them. For the none weakly totally dominated policies in condition 1), having a constant $\mathcal{U}_{\underline{\mathcal{R}}}(\pi) \equiv c$ is possible if and only if for any policy $\pi \in \Pi_{\neg wed}$, $c \in [\min\limits_{r \in R} U_r(\pi'), \max\limits_{r \in R} U_r(\pi')]$. As mentioned in the proof of Lemma 2, $c$ can exist within $[\min\limits_{r \in R} U_r(\pi), \max\limits_{r \in R} U_r(\pi)]$. Hence, $c = \max\limits_{\pi' \in \Pi_{\neg wtd}} \min\limits_{r \in R} U_r(\pi')$ is a valid assignment. $\qquad\qquad\qquad\qquad\qquad$ $\square$

Then by letting $\overline{\mathcal{R}}(\pi) := \frac{Regret(\pi, r_\pi^*)}{c - U_{r_\pi^*}(\pi)}$, we have the following theorem.

**Theorem 5.** *By letting $\mathcal{R}(\pi) := \overline{\mathcal{R}}(\pi) \cdot \delta_{r_\pi^*} + (1 - \overline{\mathcal{R}}(\pi)) \cdot \underline{\mathcal{R}}(\pi)$ with $\overline{\mathcal{R}}(\pi) := \frac{Regret(\pi, r_\pi^*)}{c - U_{r_\pi^*}(\pi)}$ and any $\underline{\mathcal{R}}$ that satisfies Lemma 4,*

$$MinimaxRegret(R) = \arg\max_{\pi \in \Pi} \mathcal{U}_\mathcal{R}(\pi) \tag{12}$$

*Proof.* If a policy $\pi \in \Pi$ is totally dominated by some other policy, since there exists another policy with larger $\mathcal{U}_\mathcal{R}$, $\pi$ cannot be a solution to $\arg\max\limits_{\pi \in \Pi} \mathcal{U}_\mathcal{R}(\pi)$. Hence, there is no need for further discussion on totally dominated policies. We discuss the none weakly totally dominated policies

and the weakly totally dominated but not totally dominated policies (shortened to "weakly totally dominated" from now on) respectively. First we expand $\arg\max_{\pi\in\Pi} \mathcal{U}_{\mathcal{R}}(\pi)$ as in Eq.13.

$$\begin{aligned}
&\arg\max_{\pi\in\Pi} \mathcal{U}_{\mathcal{R}}(\pi) \\
=\ &\arg\max_{\pi\in\Pi} \sum_{r\in R} \mathcal{R}(\pi)(r) \cdot U_r(\pi) \\
=\ &\arg\max_{\pi\in\Pi} \frac{Regret(\pi,r_\pi^*) \cdot U_{r_\pi^*}(\pi) + (\mathcal{U}_{\mathcal{R}}(\pi) - U_{r_\pi^*}(\pi) - Regret(\pi,r_\pi^*)) \cdot \mathcal{U}_{\mathcal{R}}(\pi)}{c - U_{r_\pi^*}(\pi)} \\
=\ &\arg\max_{\pi\in\Pi} \frac{(\mathcal{U}_{\mathcal{R}}(\pi) - U_{r_\pi^*}(\pi)) \cdot \mathcal{U}_{\mathcal{R}}(\pi) - (\mathcal{U}_{\mathcal{R}}(\pi) - U_{r_\pi^*}(\pi)) \cdot Regret(\pi,r_\pi^*))}{c - U_{r_\pi^*}(\pi)} \\
=\ &\arg\max_{\pi\in\Pi} \frac{\mathcal{U}_{\mathcal{R}}(\pi) - U_{r_\pi^*}(\pi)}{c - U_{r_\pi^*}(\pi)} \cdot \mathcal{U}_{\mathcal{R}}(\pi) - Regret(\pi,r_\pi^*)
\end{aligned} \tag{13}$$

1) For the none weakly totally dominated policies, since by design $\mathcal{U}_{\underline{\mathcal{R}}} \equiv c$, Eq.13 is equivalent to $\arg\max_{\pi\in\Pi_1} - Regret(\pi,r_\pi^*)$ which exactly equals $MinimaxRegret(R)$. Hence, the equivalence holds among the none weakly totally dominated policies. Furthermore, if a none weakly totally dominated policy $\pi \in \Pi_{\neg wtd}$ achieves optimality in $MinimaxRegret(R)$, its $\mathcal{U}_{\mathcal{R}}(\pi)$ is also no less than any weakly totally dominated policy. Because according to Lemma 3, for any weakly totally dominated policy $\pi_1$, its $\mathcal{U}_{\mathcal{R}}(\pi_1) \leq c$, hence $\frac{\mathcal{U}_{\underline{\mathcal{R}}}(\pi) - U_{r_\pi^*}(\pi)}{c - U_{r_\pi^*}(\pi)} \cdot \mathcal{U}_{\mathcal{R}}(\pi_1) \leq c$. Since $Regret(\pi,r_\pi^*) \leq Regret(\pi_1,r_{\pi_1}^*)$, $\mathcal{U}_{\mathcal{R}}(\pi) \geq \mathcal{U}_{\mathcal{R}}(\pi_1)$. Therefore, we can assert that if a none weakly totally dominated policy $\pi$ is a solution to $MinimaxRegret(R)$, it is also a solution to $\arg\max_{\pi\in\Pi} \mathcal{U}_{\mathcal{R}}(\pi)$. Additionally, to prove that if a none weakly totally dominated policy $\pi$ is a solution to $\arg\max_{\pi'\in\Pi} \mathcal{U}_{\mathcal{R}}(\pi')$, it is also a solution to $MinimaxRegret(R)$, it is only necessary to prove that $\pi$ achieve no larger regret than all the weakly totally dominated policies. But we delay the proof to 2).

2) If a policy $\pi$ is weakly totally dominated and is a solution to $MinimaxRegret(R)$, we show that it is also a solution to $\arg\max_{\pi\in\Pi} \mathcal{U}_{\mathcal{R}}(\pi)$, i.e., its $\mathcal{U}_{\mathcal{R}}(\pi)$ is no less than that of any other policy.

We start by comparing with non weakly totally dominated policy. for any weakly totally dominated policy $\pi_1 \in MinimaxRegret(R)$, it must hold true that $Regret(\pi_1,r_{\pi_1}^*) \leq Regret(\pi_2,r_{\pi_2}^*)$ for any $\pi_2 \in \Pi$ that weakly totally dominates $\pi_1$. However, it also holds that $Regret(\pi_2,r_{\pi_2}^*) \leq Regret(\pi_1,r_{\pi_1}^*)$ due to the weak total domination. Therefore, $Regret(\pi_1,r_{\pi_1}^*) = Regret(\pi_2,r_{\pi_2}^*) = Regret(\pi_1,r_{\pi_2}^*)$, implying that $\pi_2$ is also a solution to $MinimaxRegret(R)$. It also implies that $U_{r_{\pi_2}^*}(\pi_1) = U_{r_{\pi_2}^*}(\pi_2) \geq U_{r_{\pi_1}^*}(\pi_1)$ due to the weak total domination. However, by definition $U_{r_{\pi_1}^*}(\pi_1) \geq U_{r_{\pi_2}^*}(\pi_1)$. Hence, $U_{r_{\pi_1}^*}(\pi_1) = U_{r_{\pi_2}^*}(\pi_1) = U_{r_{\pi_2}^*}(\pi_2)$ must hold. Now we discuss two possibilities: a) there exists another policy $\pi_3$ that weakly totally dominates $\pi_2$; b) there does not exist any other policy that weakly totally dominates $\pi_2$. First, condition a) cannot hold. Because inductively it can be derived $U_{r_{\pi_1}^*}(\pi_1) = U_{r_{\pi_2}^*}(\pi_1) = U_{r_{\pi_2}^*}(\pi_2) = U_{r_{\pi_3}^*}(\pi_3)$, while Lemma 1 indicates that $\pi_3$ totally dominates $\pi_1$, which is a contradiction. Hence, there does not exist any policy that weakly totally dominates $\pi_2$, meaning that condition b) is certain. We note that $U_{r_{\pi_1}^*}(\pi_1) = U_{r_{\pi_2}^*}(\pi_1) = U_{r_{\pi_2}^*}(\pi_2)$ and the weak total domination between $\pi_1,\pi_2$ imply that $r_{\pi_1}^*, r_{\pi_2}^* \in \arg\max_{r\in R} U_r(\pi_1)$, $r_{\pi_2}^* \in \arg\min_{r\in R} U_r(\pi_2)$, and thus $\min_{r\in R} U_r(\pi_2) \leq \max_{\pi\in\Pi_{\neg wtd}} \min_{r\in R} U_r(\pi) = c$. Again, $\pi_1 \in MinimaxRegret(R)$ makes $Regret(\pi_1,r_\pi^*) \leq Regret(\pi_1,r_{\pi_1}^*) \leq Regret(\pi,r_\pi^*)$ not only hold for $\pi = \pi_2$ but also for any other policy $\pi \in \Pi_{\neg wtd}$, then for any policy $\pi \in \Pi_{\neg wtd}$, $U_{r_\pi^*}(\pi_1) \geq U_{r_\pi^*}(\pi) \geq \min_{r\in R} U_r(\pi)$. Hence, $U_{r_\pi^*}(\pi_1) \geq \max_{\pi\in\Pi_{\neg wtd}} \min_{r\in R} U_r(\pi) = c$. Since $U_{r_\pi^*}(\pi_1) = \min_{r\in R} U_r(\pi_2)$ as aforementioned, $\min_{r\in R} U_r(\pi_2) > \max_{\pi\in\Pi_{\neg wtd}} \min_{r\in R} U_r(\pi)$ will cause a contradiction. Hence, $\min_{r\in R} U_r(\pi_2) = \max_{\pi\in\Pi_{\neg wtd}} \min_{r\in R} U_r(\pi) = c$. As a result, $\mathcal{U}_{\underline{\mathcal{R}}}(\pi) = U_{r_\pi^*}(\pi) = \max_{\pi'\in\Pi_{\neg wtd}} \min_{r\in R} U_r(\pi') = c$, and $\mathcal{U}_{\mathcal{R}}(\pi) = c - Regret(\pi,r_\pi^*) \geq \max_{\pi'\in\Pi_{\neg wtd}} c - Regret(\pi',r_{\pi'}^*) = \max_{\pi'\in\Pi_{\neg wtd}} \mathcal{U}_{\mathcal{R}}(\pi')$. In other words, if a weakly

totally dominated policy $\pi$ is a solution to $MinimaxRegret(R)$, then its $\mathcal{U}_\mathcal{R}(\pi)$ is no less than that of any non weakly totally dominated policy. This also complete the proof at the end of 1), because if a none weakly totally dominated policy $\pi_1$ is a solution to $\arg\max_{\pi\in\Pi}\mathcal{U}_\mathcal{R}(\pi)$ but not a solution to $MinimaxRegret(R)$, then $Regret(\pi_1, r_{\pi_1}^*) > 0$ and a weakly totally dominated policy $\pi_2$ must be the solution to $MinimaxRegret(R)$. Then, $\mathcal{U}_\mathcal{R}(\pi_2) = c > c - Regret(\pi_1, r_{\pi_1}^*) = \mathcal{U}_\mathcal{R}(\pi_1)$, which, however, contradicts $\pi_1 \in \arg\max_{\pi\in\Pi}\mathcal{U}_\mathcal{R}(\pi)$.

It is obvious that a weakly totally dominated policy $\pi \in MinimaxRegret(R)$ has a $\mathcal{U}_\mathcal{R}(\pi)$ no less than any other weakly totally dominated policy. Because for any other weakly totally dominated policy $\pi_1$, $\mathcal{U}_{\underline{\mathcal{R}}}(\pi_1) \leq c$ and $Regret(\pi_1, r_{\pi_1}^*) \leq Regret(\pi, r_\pi^*)$, hence $\mathcal{U}_\mathcal{R}(\pi_1) \leq \mathcal{U}_\mathcal{R}(\pi)$ according to Eq.13.

So far we have shown that if a weakly totally dominated policy $\pi$ is a solution to $MinimaxRegret(R)$, it is also a solution to $\arg\max_{\pi'\in\Pi}\mathcal{U}_\mathcal{R}(\pi')$. Next, we need to show that the reverse is also true, i.e., if a weakly totally dominated policy $\pi$ is a solution to $\arg\max_{\pi\in\Pi}\mathcal{U}_\mathcal{R}(\pi)$, it must also be a solution to $MinimaxRegret(R)$. In order to prove its truthfulness, we need to show that if $\pi \notin MinimaxRegret(R)$, whether there exists: a) a none weakly totally dominated policy $\pi_1$, or b) another weakly totally dominated policy $\pi_1$, such that $\pi_1 \in MinimaxRegret(R)$ and $\mathcal{U}_\mathcal{R}(\pi_1) \leq \mathcal{U}_\mathcal{R}(\pi)$. If neither of the two policies exists, we can complete our proof. Since it has been proved in 1) that if a none weakly totally dominated policy achieves $MinimaxRegret(R)$, it also achieves $\arg\max_{\pi'\in\Pi}\mathcal{U}_\mathcal{R}(\pi')$, the policy described in condition a) does not exist. Hence, it is only necessary to prove that the policy in condition b) also does not exist.

If such weakly totally dominated policy $\pi_1$ exists, $\pi \notin MinimaxRegret(R)$ and $\pi_1 \in MinimaxRegret(R)$ indicates $Regret(\pi, r_\pi^*) > Regret(\pi_1, r_{\pi_1}^*)$. Since $\mathcal{U}_\mathcal{R}(\pi_1) \geq \mathcal{U}_\mathcal{R}(\pi)$, according to Eq.13, $\mathcal{U}_\mathcal{R}(\pi_1) = c - Regret(\pi_1, r_{\pi_1}^*) \leq \mathcal{U}_\mathcal{R}(\pi) = \frac{\mathcal{U}_{\underline{\mathcal{R}}}(\pi) - U_{r_\pi^*}(\pi)}{c - U_{r_\pi^*}(\pi)} \cdot \mathcal{U}_{\underline{\mathcal{R}}}(\pi) - Regret(\pi, r_\pi^*)$. Thus $\frac{\mathcal{U}_{\underline{\mathcal{R}}}(\pi) - U_{r_\pi^*}(\pi)}{c - U_{r_\pi^*}(\pi)}(\pi) \cdot \mathcal{U}_{\underline{\mathcal{R}}} \geq c + Regret(\pi, r_\pi^*) - Regret(\pi_1, r_{\pi_1}^*) > c$, which is impossible due to $\mathcal{U}_{\underline{\mathcal{R}}} \leq c$. Therefore, such $\pi_1$ also does not exist. In fact, this can be reasoned from another perspective. If there exists a weakly totally dominated policy $\pi_1$ with $U_{r_{\pi_1}^*}(\pi_1) = c = U_{r_\pi^*}(\pi)$ but $\pi_1 \notin MinimaxRegret(R)$, then $Regret(\pi, r_\pi^*) > Regret(\pi_1, r_{\pi_1}^*)$. It also indicates $\max_{\pi'\in\Pi} U_{r_\pi^*}(\pi') > \max_{\pi'\in\Pi} U_{r_{\pi_1}^*}(\pi')$. Meanwhile, $Regret(\pi_1, r_\pi^*) := \max_{\pi'\in\Pi} U_{r_\pi^*}(\pi') - U_{r_\pi^*}(\pi_1) \leq Regret(\pi_1, r_{\pi_1}^*) := \max_{\pi'\in\Pi} U_{r_{\pi_1}^*}(\pi') - U_{r_{\pi_1}^*}(\pi_1) := \max_{r\in R}\max_{\pi'\in\Pi} U_r(\pi') - U_r(\pi_1)$ indicates $\max_{\pi'\in\Pi} U_{r_\pi^*}(\pi') - \max_{\pi'\in\Pi} U_{r_{\pi_1}^*}(\pi') \leq U_{r_\pi^*}(\pi_1) - U_{r_{\pi_1}^*}(\pi_1)$. However, we have proved that, for a weakly totally dominated policy, $\pi_1 \in MinimaxRegret(R)$ indicates $U_{r_{\pi_1}^*}(\pi_1) = \max_{r\in R} U_r(\pi_1)$. Hence, $\max_{\pi'\in\Pi} U_{r_\pi^*}(\pi') - \max_{\pi'\in\Pi} U_{r_{\pi_1}^*}(\pi') \leq U_{r_\pi^*}(\pi_1) - U_{r_{\pi_1}^*}(\pi_1) \leq 0$ and it contradicts $\max_{\pi'\in\Pi} U_{r_\pi^*}(\pi') > \max_{\pi'\in\Pi} U_{r_{\pi_1}^*}(\pi')$. Therefore, such $\pi_1$ does not exist. In summary, we have exhausted all conditions and can assert that for any policies, being a solution to $MinimaxRegret(R)$ is equivalent to a solution to $\arg\max_{\pi\in\Pi}\mathcal{U}_\mathcal{R}(\pi)$. We complete our proof.

$\square$

## A.3  MEASURING POLICY PERFORMANCE

Recall that the function $U_r(\pi)$ is used to measure the performance of a policy $\pi$ under a reward function $r$. In Dennis et al. (2020), $U_r(\pi) = \eta_r(\pi)$. In this section, we discuss the validity of letting $U_r(\pi)$ be the loss function of a generic IRL objective, e.g., $U_r(\pi) = \eta_r(\pi) - \eta_r(\pi_E)$ where $\eta_r(\pi_E)$ measures the expected return of the expert policy $\pi_E$ and can be estimated if an expert demonstration set $E$ instead of $\pi_E$ is provided. If further letting $R_E = \{r | r \in \arg\min_{r'\in R}\max_{\pi\in\Pi} U_{r'}(\pi) - U_{r'}(\pi_E)\}$, $\max_{\pi\in\Pi} U_r(\pi)$ is a constant for any $r \in R_E$, notated as $u := \max_{\pi\in\Pi} U_r(\pi)$. Because by definition $R_E = \{r | r \in \arg\min_{r\in R}\max_{\pi\in\Pi} U_r(\pi)\}$. If there exists $r_1, r_2 \in R_E$ such that $\max_{\pi\in\Pi} U_{r_1}(\pi) < \max_{\pi\in\Pi} U_{r_2}(\pi)$, $r_2$ will not be a member of $R_E$. Furthermore, $\{U_r(\pi) | \pi \in \Pi, r \in R_E\}$ will be upper-bounded by a

constant $u = \max_{\pi \in \Pi} U_r(\pi)$. Because if there exists a policy $\pi \in \Pi$ and a reward function $r \in R_E$ with $U_r(\pi) > u$, it contradicts the fact that $u = \max_{\pi' \in \Pi} U_r(\pi')$. In this case, $MinimaxRegret(R_E) = \arg\min_{\pi \in \Pi} \max_{r \in R_E} Regret(\pi, r) = \arg\min_{\pi \in \Pi} \max_{r \in R_E} u - U_r(\pi) = \arg\max_{\pi \in \Pi} \min_{r \in R_E} U_r(\pi)$. Note that before making any other assumption on $R_E, \Pi$ and $U_r(\cdot)$, $\max_{\pi \in \Pi} \min_{r \in R_E} U_r(\pi)$ cannot be regarded as the same as IRL itself $\min_{r \in R} \max_{\pi \in \Pi} U_r(\pi)$. The solution to $\arg\max_{\pi \in \Pi} \min_{r \in R_E} U_r(\pi)$ is the policy with the highest worst case $U_r(\pi)$ for $r \in R_E$. The IRL problem, however, may induce a policy that maximizes $U_r(\cdot)$ for some $r \in R_E$ while minimizing $U_{r'}(\cdot)$ for some other $r' \in R_E$. While $\min_{r \in R} \max_{\pi \in \Pi} U_r(\pi) = u$, it is possible that $\max_{\pi \in \Pi} \min_{r \in R_E} U_r(\pi) < u$. In fact, it is easily observable that the solutions to $MinimaxRegret$ with some $U_r(\pi)$ will be the same as that of letting $U_r(\pi) := U_r(\pi) - U_r(\pi_E)$. Hence, in this paper we simply use $\eta_r(\pi)$ as $U_r(\pi)$.

**Lemma 5.** *If a policy $\pi \in MinimaxRegret(R_E)$ when the policy performance is measured with some $U_r$, then $\pi \in MinimaxRegret(R_E)$ when letting $U_r(\pi) := U_r(\pi) - U_r(\pi_E)$.*

*Proof.* When using $U_r(\pi) := U_r(\pi) - U_r(\pi_E)$ to measure the policy performance, solving $MinimaxRegret(R)$ is to solve Eq. 14, which is the same as Eq.9.

$$
\begin{aligned}
MimimaxRegret(R_E) &= \arg\max_{\pi \in \Pi} \min_{r \in R_E} Regret(\pi, r) \\
&= \arg\max_{\pi \in \Pi} \min_{r \in R_E} \max_{\pi' \in \Pi} \{U_r(\pi') - U_r(\pi_E)\} - (U_r(\pi) - U_r(\pi_E)) \\
&= \arg\max_{\pi \in \Pi} \min_{r \in R_E} \max_{\pi' \in \Pi} U_r(\pi') - U_r(\pi) \quad (14)
\end{aligned}
$$

$\square$

## A.4 Criterion for Successful Policy Learning

**Theorem 1.**(Task-Failure Avoidance) If the following conditions (1) (2) hold for $R$, then the optimal protagonist policy $\pi_P := MinimaxRegret(R)$ satisfies that $\forall r_{al} \in R, U_{r_{al}}(\pi_P) \notin F_{r_{al}}$.

(1) There exists $r_{al} \in R$, and $\max_{r_{al} \in R} \{\sup F_{r_{al}} - \inf F_{r_{al}}\} < \min_{r_{al} \in R} \{\inf S_{r_{al}} - \sup F_{r_{al}}\} \wedge$ $\max_{r_{al} \in R} \{\sup S_{r_{al}} - \inf S_{r_{al}}\} < \min_{r_{al} \in R} \{\inf S_{r_{al}} - \sup F_{r_{al}}\}$;

(2) There exists a policy $\pi^*$ such that $\forall r_{al} \in R, U_{r_{al}}(\pi^*) \in S_{r_{al}}$, and $\forall r_{mis} \in R$, $\max_{\pi \in \Pi} U_{r_{mis}}(\pi) - U_{r_{mis}}(\pi^*) < \min_{r_{al} \in R} \{\inf S_{r_{al}} - \sup F_{r_{al}}\}$.

*Proof.* Suppose the conditions are met, and a policy $\pi_1$ satisfies the property described in conditions 2). Then for any policy $\pi_2 \in MinimaxRegret(R)$, if $\pi_2$ does not satisfy the mentioned property, there exists a task-aligned reward function $r_{al} \in R_E$ such that $U_{r_{al}}(\pi_2) \in F_{r_{al}}$. In this case $Regret(\pi_2, r_{al}) = \max_{\pi \in \Pi} U_{r_{al}}(\pi) - U_{r_{al}}(\pi_2) \geq \inf S_{r_{al}} - \sup F_{r_{al}} \geq \min_{r'_{al} \in R_E} \{\inf S_{r'_{al}} - \sup F_{r'_{al}}\}$. However, for $\pi_1$, it holds for any task-aligned reward function $\hat{r}_{al} \in R_E$ that $Regret(\pi_2, \hat{r}_{al}) \leq \sup S_{\hat{r}_{al}} - \inf S_{\hat{r}_{al}} < \min_{r'_{al} \in R_E} \{\inf S_{r'_{al}} - \sup F_{r'_{al}}\}$, and it also holds for any misaligned reward function $r_{mis} \in R_E$ that $Regret(\pi_2, r_{mis}) = \max_{\pi \in \Pi} U_{r_{mis}}(\pi) - U_{r_{mis}}(\pi_2) < \min_{r'_{al} \in R_E} \{\inf S_{r_{al}} - \sup F_{r_{al}}\}$. Hence, $Regret(\pi_2, r_{al}) < Regret(\pi_1, r_{al})$, contradicting $\pi_1 \in MiniRegret$. We complete the proof. $\square$

**Corollary 1.**(Task-Success Guarantee) Assume that Condition (1) in Theorem 1 is satisfied. If there exists a policy $\pi^*$ such that $\forall r \in R, \max_{\pi} U_r(\pi) - U_r(\pi^*) < \min_{r_{al} \in R} \{\sup S_{r_{al}} - \inf S_{r_{al}}\}$, then the optimal protagonist policy $\pi_P := MinimaxRegret(R)$ satisfies that $\forall r_{al} \in R, U_{r_{al}}(\pi) \in S_{r_{al}}$.

*Proof.* Since $\underset{r\in R}{max}\ \underset{\pi}{\max}\ U_r(\pi) - U_r(\pi_P) \leq \underset{r\in R}{max}\ \underset{\pi}{\max}\ U_r(\pi) - U_r(\pi^*) < \underset{r_{al}\in R}{\min}\ \{\sup S_{r_{al}} - \inf S_{r_{al}}\}$, we can conclude that for any $r_{al} \in R$, $\underset{\pi}{\max}\ U_{r_{al}}(\pi) - U_{r_{al}}(\pi_P) \leq \{\sup S_{r_{al}} - \inf S_{r_{al}}$, in other words, $U_{r_{al}}(\pi_P) \in S_{r_{al}}$. The proof is complete. $\qquad\square$

**Theorem 2.**)(Task-Failure Avoidance) If the following conditions (1) (2) hold for $R_{E,\delta}$, then the optimal protagonist policy $\pi_P := MinimaxRegret(R_{E,\delta})$ satisfies that $\forall r_{al} \in R_{E,\delta}$, $U_{r_{al}}(\pi) \notin F_{r_{al}}$.

    (1) The condition (1) in Theorem 1 holds

    (2) $\forall r_{al} \in R_{E,\delta}$, $L_{r_{al}} \cdot W_E - \delta \leq \sup S_{r_{al}} - \inf S_{r_{al}}$ and $\forall r_{mis} \in R_{E,\delta}$, $L_{r_{mis}} \cdot W_E - \delta < \underset{r_{al}\in R_{E,\delta}}{\min}\ \{\inf S_{r_{al}} - \sup F_{r_{al}}\}$.

*Proof.* We consider $U_r(\pi) = \mathbb{E}_{\tau\sim\pi}[r(\tau)]$. Since $W_E \triangleq \underset{\pi\in\Pi}{\min}\ W_1(\pi, E) = \frac{1}{K} \underset{\|r\|_L \leq K}{\sup}\ U_r(E) - U_r(\pi)$ for any $K > 0$, let $\pi^*$ be the policy that achieves the minimality in $W_E$. Then for any $r_{al} \in R$, the term $L_{r_{al}} \cdot W_E - \delta \geq L_{r_{al}} \cdot \frac{1}{L_{r_{al}}} \underset{\|r\|_L \leq L_{r_{al}}}{\sup}\ U_r(E) - U_r(\pi) \geq U_{r_{al}}(E) - U_{r_{al}}(\pi)$.

Hence, for all $r_{al} \in R$, $U_{r_{al}}(E) - U_{r_{al}}(\pi) < \sup S_{r_{al}} - \inf S_{r_{al}}$, i.e., $U_{r_{al}}(\pi^*) \in S_{r_{al}}$. Likewise, $L_{r_{mis}} \cdot W_E - \delta < \underset{r_{al}\in R_{E,\delta}}{\min}\ \{\inf S_{r_{al}} - \sup F_{r_{al}}\}$ indicates that for all $r_{mis} \in R$, $U_{r_{mis}}(E) - U_{r_{mis}}(\pi) < \underset{r_{al}\in R_{E,\delta}}{\min}\ \{\inf S_{r_{al}} - \sup F_{r_{al}}\}$. Then, we have recovered the condition (2) in Theorem 1. As a result, we deliver the same guarantees in Theorem 1. $\qquad\square$

**Corollary 2.**(Task-Success Guarantee) Assume that the condition (1) in Theorem 1 holds for $R_{E,\delta}$. If for any $r \in R_{E,\delta}$, $L_r \cdot W_E - \delta \leq \underset{r_{al}\in R}{\min}\ \{\sup S_{r_{al}} - \inf S_{r_{al}}\}$, then the optimal protagonist policy $\pi_P = MinimaxRegret(R_{E,\delta})$ satisfies $\forall r_{al} \in R_{E,\delta}$, $U_{r_{al}}(\pi) \in S_{r_{al}}$.

*Proof.* Again, we let $\pi^*$ be the policy that achieves the minimality in $W_E$. Then, we have $L_r \cdot W_E - \delta \geq L_r \cdot \frac{1}{L_r} \underset{\|r\|_L \leq L_r}{\sup}\ U_r(E) - U_r(\pi^*) \geq U_r(E) - U_r(\pi^*)$ for any $r \in R$. We have recovered the condition in Corollary 1. The proof is complete. $\qquad\square$

**Theorem3.** Let the IRL loss be in the form of $J_{IRL}(\pi, r) := U_r(\pi) - U_r(\pi_E)$ for some $U_r(\pi)$. If $\arg\underset{r\in R}{\min}\ \underset{\pi\in\Pi}{\max}\ J_{IRL}(r_E, \pi)$ can reach Nash Equilibrium with a reward function set $R_E$ and a policy set $\Pi_E$, then $\Pi_E$ equals the set of solutions to $MinimiaxRegret$.

*Proof.* The reward function set $R_E$ and the policy set $\Pi_E$ achieving Nash Equilibrium for $\arg\underset{r\in R}{\min}\ \underset{\pi\in\Pi}{\max}\ J_{IRL}(r_E, \pi)$ indicates that for any $r \in R_E, \pi \in \Pi_E$, $\pi \in \arg\underset{\pi\in\Pi}{\max}\ U_r(\pi) - U_r(\pi_E)$. Then $\Pi_E$ will be the solution to $\arg\underset{\pi_P\in\Pi}{\max}\ \underset{r\in R_E}{\min}\ \left\{ \underset{\pi_A\in\Pi}{\max}\ U_r(\pi_A) - U_r(\pi_E) \right\} - (U_r(\pi_P) - U_r(\pi_E))$ because the policies in $\Pi_E$ achieve zero regret. Then Lemma 5 states that $\Pi_E$ will also be the solution to $\arg\underset{\pi_P\in\Pi}{\max}\ \underset{r\in R_E}{\min}\ \left\{ \underset{\pi_A\in\Pi}{\max}\ U_r(\pi_A) \right\} - U_r(\pi_P)$. We finish the proof. $\qquad\square$

A.5   RUNNING EXAMPLE

**Example** 1 In a two-state transition system where $s_0$ is the initial state, and $s_1$ is an absorbing state, an agent can choose action $a_0$ at state $s_0$ to stay at $s_0$ or choose $a_1$ to reach $s_1$. Any agent can start from $s_0$ and choose actions for 5 timesteps. The task is to learn a stochastic policy to reach $s_1$. Expert only demonstrates one trajectory $E = \{\tau_E = (s_0, a_1, s_1, s_1, s_1, s_1)\}$, i.e., choose $a_1$ at $s_0$ and then stay in $s_1$ for the rest 4 steps. The convex combinations of two basis functions $r_1$ and $r_2$ constitute the hypothesis set of reward functions $R = \{r|r(s,a) = \alpha \cdot r_1(s,a) + (1-\alpha) \cdot r_2(s,a), \alpha \in [0,1]\}$ where $r_1(s,a) \equiv 4$ constantly, and $r_2(s_0, a_0) = r_2(s_0, a_1) = 0, r_2(s_1, \cdot) = 5$. We now prove that when applying IRL to learn a reward function from $E$, any convex combination of $r_1$ and $r_2$ is an

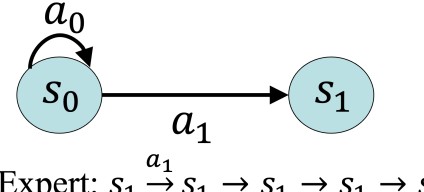

Expert: $s_1 \xrightarrow{a_1} s_1 \rightarrow s_1 \rightarrow s_1 \rightarrow s_1$

|        | $(s_0, a_0)$ | $(s_0, a_1)$ | $(s_1, \cdot)$ |
|--------|--------------|--------------|----------------|
| $r_1$  | 4            | 4            | 4              |
| $r_2$  | 0            | 0            | 5              |

Figure 6: ss

optimal solution, i.e., $R_E = R$. Furthermore, we prove that $MinimaxRegret(R)$ will produce $\pi_P(a_1|s_0) = 1$ which is the desired solution.

*Proof.* For any policy $\pi$ and reward function $r \in R$, the policy performance $U_r(\pi)$ can be represented as follows.

$$
\begin{aligned}
U_r(\pi) &= 4\alpha\pi(a_0|s_0)^t + \sum_{t=0}^{t}(4\alpha t + 4\alpha + (4\alpha + 5(1-\alpha))(4-t)) \cdot \pi(a_0|s_0)^t(1 - \pi(a_0|s_0)) \\
&= 20 - 5(1-\alpha)\sum_{t=1}^{4}\pi(a_0|s_0)^4 \tag{15}
\end{aligned}
$$

Then for any $\alpha \in [0,1)$, $\max_{\pi} U_r(\pi) = 20$, and the optimal policy is $\pi^*(a_0|s_0) = 0, \pi^*(a_1|s_0) = 1$. For $\alpha = 1$, $U_r(\pi) = 20$ constantly for any $\pi$. For expert demonstration $E$, $U_r(E) = 4\alpha \cdot 5 + 5(1-\alpha) \cdot 4 \equiv 20$. Therefore, any $r = \alpha r_1 + (1-\alpha)r_2$ with $\alpha \in [0,1]$ can be the optimal solution to IRL since they all induce the same performance gap $U_r(E) - \max_{\pi} U_r(\pi) = 0$. This ambiguity in choosing reward functions can cause reward misalignment because the reward function $r = r_1$ induced by $\alpha = 1$ violates the criterion of task-reward alignment in Definition 1: achieving optimality under $r_1$ does not guarantee task success since all policies are optimal under $r_1$. On the other hand, $MinimaxRegret(R)$ induces a policy $\pi_P(a_1|s_0) = 1$. Because this policy has the lowest worst-case regret $Regret(\pi_P, r) = \max_{r \in R} \max_{\pi_A} U_r(\pi_A) - U_r(\pi_P)) = 0$. $\qquad\square$

## B  APPROACH TO SOLVING MINIMAXREGRET

In this section, we develop a series of theories that lead to two bounds of the Protagonist Antagonist Induced Regret. By using those bounds, we formulate objective functions for solving Imitation Learning problems with PAGAR.

### B.1  PROTAGONIST ANTAGONIST INDUCED REGRET BOUNDS

Our theories are inspired by the on-policy policy improvement methods in Schulman et al. (2015). The theories in Schulman et al. (2015) are under the setting where entropy regularizer is not considered. In our implementation, we always consider entropy regularized RL of which the objective is to learn a policy that maximizes $J_{RL}(\pi; r) = \eta_r(\pi) + \mathcal{H}(\pi)$. Also, since we use GAN-based IRL algorithms, the learned reward function $r$ as proved by Fu et al. (2018) is a distribution. Moreover, it is also proved in Fu et al. (2018) that a policy $\pi$ being optimal under $r$ indicates that $\log \pi \equiv r \equiv \mathcal{A}_\pi$. We omit the proof and let the reader refer to Fu et al. (2018) for details. Although all our theories are about the relationship between the Protagonist Antagonist Induced Regret and the soft advantage function $\mathcal{A}_\pi$, the equivalence between $\mathcal{A}_\pi$ and $r$ allows us to use the theories to formulate our

reward optimization objective functions. To start off, we denote the reward function to be optimized as $r$. Given the intermediate learned reward function $r$, we study the Protagonist Antagonist Induced Regret between two policies $\pi_1$ and $\pi_2$.

**Lemma 6.** *Given a reward function $r$ and a pair of policies $\pi_1$ and $\pi_2$,*

$$\eta_r(\pi_1) - \eta_r(\pi_2) = \mathop{\mathbb{E}}_{\tau \sim \pi_1} \left[ \sum_{t=0}^{\infty} \gamma^t \mathcal{A}_{\pi_2}(s^{(t)}, a^{(t)}) \right] + \mathop{\mathbb{E}}_{\tau \sim \pi} \left[ \sum_{t=0}^{\infty} \gamma^t \mathcal{H} \left( \pi_2(\cdot | s^{(t)}) \right) \right] \tag{16}$$

*Proof.* This proof follows the proof of Lemma 1 in Schulman et al. (2015) where RL is not entropy-regularized. For entropy-regularized RL, since $\mathcal{A}_\pi(s, a^{(t)}) = \mathop{\mathbb{E}}_{s' \sim \mathcal{P}(\cdot | s, a^{(t)})} \left[ r(s, a^{(t)}) + \gamma \mathcal{V}_\pi(s') - \mathcal{V}_\pi(s) \right]$,

$$
\begin{aligned}
& \mathop{\mathbb{E}}_{\tau \sim \pi_1} \left[ \sum_{t=0}^{\infty} \gamma^t \mathcal{A}_{\pi_2}(s^{(t)}, a^{(t)}) \right] \\
= \;& \mathop{\mathbb{E}}_{\tau \sim \pi_1} \left[ \sum_{t=0}^{\infty} \gamma^t \left( r(s^{(t+1)}, a^{(t+1)}) + \gamma \mathcal{V}_{\pi_2}(s^{(t+1)}) - \mathcal{V}_{\pi_2}(s^{(t)}) \right) \right] \\
= \;& \mathop{\mathbb{E}}_{\tau \sim \pi_1} \left[ \sum_{t=0}^{\infty} \gamma^t r(s^{(t)}, a^{(t)}) - \mathcal{V}_{\pi_2}(s^{(0)}) \right] \\
= \;& \mathop{\mathbb{E}}_{\tau \sim \pi_1} \left[ \sum_{t=0}^{\infty} \gamma^t r(s^{(t)}, a^{(t)}) \right] - \mathop{\mathbb{E}}_{s^{(0)} \sim d_0} \left[ \mathcal{V}_{\pi_2}(s^{(0)}) \right] \\
= \;& \mathop{\mathbb{E}}_{\tau \sim \pi_1} \left[ \sum_{t=0}^{\infty} \gamma^t r(s^{(t)}, a^{(t)}) \right] - \mathop{\mathbb{E}}_{\tau \sim \pi_2} \left[ \sum_{t=0}^{\infty} \gamma^t r(s^{(t)}, a^{(t)}) + \mathcal{H} \left( \pi_2(\cdot | s^{(t)}) \right) \right] \\
= \;& \eta_r(\pi_1) - \eta_r(\pi_2) - \mathop{\mathbb{E}}_{\tau \sim \pi_2} \left[ \sum_{t=0}^{\infty} \gamma^t \mathcal{H} \left( \pi_2(\cdot | s^{(t)}) \right) \right] \\
= \;& \eta_r(\pi_1) - \eta_r(\pi_2) - \mathcal{H}(\pi_2)
\end{aligned}
$$

$\square$

**Remark 1.** Lemma 6 confirms that $\mathop{\mathbb{E}}_{\tau \sim \pi} \left[ \sum_{t=0}^{\infty} \gamma^t \mathcal{A}_\pi(s^{(t)}, a^{(t)}) \right] = \eta_r(\pi) - \eta_r(\pi) + \mathcal{H}(\pi) = \mathcal{H}(\pi)$.

We follow Schulman et al. (2015) and denote $\Delta \mathcal{A}(s) = \mathop{\mathbb{E}}_{a \sim \pi_1(\cdot | s)} [\mathcal{A}_{\pi_2}(s, a)] - \mathop{\mathbb{E}}_{a \sim \pi_2(\cdot | s)} [\mathcal{A}_{\pi_2}(s, a)]$ as the difference between the expected advantages of following $\pi_2$ after choosing an action respectively by following policy $\pi_1$ and $\pi_2$ at any state $s$. Although the setting of Schulman et al. (2015) differs from ours by having the expected advantage $\mathop{\mathbb{E}}_{a \sim \pi_2(\cdot | s)} [\mathcal{A}_{\pi_2}(s, a)]$ equal to 0 due to the absence of entropy regularization, the following definition and lemmas from Schulman et al. (2015) remain valid in our setting.

**Definition 7.** Schulman et al. (2015), the protagonist policy $\pi_1$ and the antagonist policy $\pi_2$) are $\alpha$-coupled if they defines a joint distribution over $(a, \tilde{a}) \in \mathbb{A} \times \mathbb{A}$, such that $Prob(a \neq \tilde{a} | s) \leq \alpha$ for all $s$.

**Lemma 7.** *Schulman et al. (2015) Given that the protagonist policy $\pi_1$ and the antagonist policy $\pi_2$ are $\alpha$-coupled, then for all state $s$,*

$$|\Delta \mathcal{A}(s)| \leq 2\alpha \max_a |\mathcal{A}_{\pi_2}(s, a)| \tag{17}$$

**Lemma 8.** *Schulman et al. (2015) Given that the protagonist policy $\pi_1$ and the antagonist policy $\pi_2$ are $\alpha$-coupled, then*

$$\left| \mathop{\mathbb{E}}_{s^{(t)} \sim \pi_1} \left[ \Delta \mathcal{A}(s^{(t)}) \right] - \mathop{\mathbb{E}}_{s^{(t)} \sim \pi_2} \left[ \Delta \mathcal{A}(s^{(t)}) \right] \right| \leq 4\alpha(1 - (1 - \alpha)^t) \max_{s,a} |\mathcal{A}_{\pi_2}(s, a)| \tag{18}$$

**Lemma 9.** *Given that the protagonist policy $\pi_1$ and the antagonist policy $\pi_2$ are $\alpha$-coupled, then*

$$\mathop{\mathbb{E}}_{\substack{s^{(t)}\sim\pi_1\\a^{(t)}\sim\pi_2}}\left[\mathcal{A}_{\pi_2}(s^{(t)},a^{(t)})\right]-\mathop{\mathbb{E}}_{\substack{s^{(t)}\sim\pi_2\\a^{(t)}\sim\pi_2}}\left[\mathcal{A}_{\pi_2}(s^{(t)},a^{(t)})\right]\leq 2(1-(1-\alpha)^t)\max_{(s,a)}|\mathcal{A}_{\pi_2}(s,a)| \quad (19)$$

*Proof.* The proof is similar to that of Lemma 8 in Schulman et al. (2015). Let $n_t$ be the number of times that $a^{(t')}\sim\pi_1$ does not equal $a^{(t')}\sim\pi_2$ for $t'<t$, i.e., the number of times that $\pi_1$ and $\pi_2$ disagree before timestep $t$. Then for $s^{(t)}\sim\pi_1$, we have the following.

$$\mathop{\mathbb{E}}_{s^{(t)}\sim\pi_1}\left[\mathop{\mathbb{E}}_{a^{(t)}\sim\pi_2}\left[\mathcal{A}_{\pi_2}(s^{(t)},a^{(t)})\right]\right]$$
$$=\ P(n_t=0)\mathop{\mathbb{E}}_{\substack{s^{(t)}\sim\pi_1\\n_t=0}}\left[\mathop{\mathbb{E}}_{a^{(t)}\sim\pi_2}\left[\mathcal{A}_{\pi_2}(s^{(t)},a^{(t)})\right]\right]+P(n_t>0)\mathop{\mathbb{E}}_{\substack{s^{(t)}\sim\pi_1\\n_t>0}}\left[\mathop{\mathbb{E}}_{a^{(t)}\sim\pi_2}\left[\mathcal{A}_{\pi_2}(s^{(t)},a^{(t)})\right]\right]$$

The expectation decomposes similarly for $s^{(t)}\sim\pi_2$.

$$\mathop{\mathbb{E}}_{\substack{s^{(t)}\sim\pi_2\\a^{(t)}\sim\pi_2}}\left[\mathcal{A}_{\pi_2}(s^{(t)},a^{(t)})\right]$$
$$=\ P(n_t=0)\mathop{\mathbb{E}}_{\substack{s^{(t)}\sim\pi_2\\a^{(t)}\sim\pi_2\\n_t=0}}\left[\mathcal{A}_{\pi_2}(s^{(t)},a^{(t)})\right]+P(n_t>0)\mathop{\mathbb{E}}_{\substack{s^{(t)}\sim\pi_2\\a^{(t)}\sim\pi_2\\n_t>0}}\left[\mathcal{A}_{\pi_2}(s^{(t)},a^{(t)})\right]$$

When computing $\mathop{\mathbb{E}}_{s^{(t)}\sim\pi_1}\left[\mathop{\mathbb{E}}_{a^{(t)}\sim\pi_2}\left[\mathcal{A}_{\pi_2}(s^{(t)},a^{(t)})\right]\right]-\mathop{\mathbb{E}}_{\substack{s^{(t)}\sim\pi_2\\a^{(t)}\sim\pi_2}}\left[\mathcal{A}_{\pi_2}(s^{(t)},a^{(t)})\right]$, the terms with $n_t=0$ cancel each other because $n_t=0$ indicates that $\pi_1$ and $\pi_2$ agreed on all timesteps less than $t$. That leads to the following.

$$\mathop{\mathbb{E}}_{s^{(t)}\sim\pi_1}\left[\mathop{\mathbb{E}}_{a^{(t)}\sim\pi_2}\left[\mathcal{A}_{\pi_2}(s^{(t)},a^{(t)})\right]\right]-\mathop{\mathbb{E}}_{\substack{s^{(t)}\sim\pi_2\\a^{(t)}\sim\pi_2}}\left[\mathcal{A}_{\pi_2}(s^{(t)},a^{(t)})\right]$$
$$=\ P(n_t>0)\mathop{\mathbb{E}}_{\substack{s^{(t)}\sim\pi_1\\n_t>0}}\left[\mathop{\mathbb{E}}_{a^{(t)}\sim\pi_2}\left[\mathcal{A}_{\pi_2}(s^{(t)},a^{(t)})\right]\right]-P(n_t>0)\mathop{\mathbb{E}}_{\substack{s^{(t)}\sim\pi_2\\a^{(t)}\sim\pi_2\\n_t>0}}\left[\mathcal{A}_{\pi_2}(s^{(t)},a^{(t)})\right]$$

By definition of $\alpha$, the probability of $\pi_1$ and $\pi_2$ agreeing at timestep $t'$ is no less than $1-\alpha$. Hence, $P(n_t>0)\leq 1-(1-\alpha^t)^t$. Hence, we have the following bound.

$$\left|\mathop{\mathbb{E}}_{s^{(t)}\sim\pi_1}\left[\mathop{\mathbb{E}}_{a^{(t)}\sim\pi_2}\left[\mathcal{A}_{\pi_2}(s^{(t)},a^{(t)})\right]\right]-\mathop{\mathbb{E}}_{\substack{s^{(t)}\sim\pi_2\\a^{(t)}\sim\pi_2}}\left[\mathcal{A}_{\pi_2}(s^{(t)},a^{(t)})\right]\right|$$
$$=\ \left|P(n_t>0)\mathop{\mathbb{E}}_{\substack{s^{(t)}\sim\pi_1\\n_t>0}}\left[\mathop{\mathbb{E}}_{a^{(t)}\sim\pi_2}\left[\mathcal{A}_{\pi_2}(s^{(t)},a^{(t)})\right]\right]-P(n_t>0)\mathop{\mathbb{E}}_{\substack{s^{(t)}\sim\pi_2\\a^{(t)}\sim\pi_2\\n_t>0}}\left[\mathcal{A}_{\pi_2}(s^{(t)},a^{(t)})\right]\right|$$
$$\leq\ P(n_t>0)\left(\left|\mathop{\mathbb{E}}_{\substack{s^{(t)}\sim\pi_1\\a^{(t)}\sim\pi_2\\n_t\geq0}}\left[\mathcal{A}_{\pi_2}(s^{(t)},a^{(t)})\right]\right|+\left|\mathop{\mathbb{E}}_{\substack{s^{(t)}\sim\pi_2\\a^{(t)}\sim\pi_2\\n_t>0}}\left[\mathcal{A}_{\pi_2}(s^{(t)},a^{(t)})\right]\right|\right)$$
$$\leq\ 2(1-(1-\alpha)^t)\max_{(s,a)}|\mathcal{A}_{\pi_2}(s,a)| \quad (20)$$

$\square$

The preceding lemmas lead to the proof for Theorem 4 in the main text.

**Theorem 4.** Suppose that $\pi_2$ is the optimal policy in terms of entropy regularized RL under $r$. Let $\alpha = \max_s D_{TV}(\pi_1(\cdot|s), \pi_2(\cdot|s))$, $\epsilon = \max_{s,a} |\mathcal{A}_{\pi_2}(s, a^{(t)})|$, and $\Delta \mathcal{A}(s) = \mathbb{E}_{a \sim \pi_1}[\mathcal{A}_{\pi_2}(s, a)] - \mathbb{E}_{a \sim \pi_2}[\mathcal{A}_{\pi_2}(s, a)]$. For any policy $\pi_1$, the following bounds hold.

$$\left| \eta_r(\pi_1) - \eta_r(\pi_2) - \sum_{t=0}^{\infty} \gamma^t \mathop{\mathbb{E}}_{s^{(t)} \sim \pi_1} \left[ \Delta \mathcal{A}(s^{(t)}) \right] \right| \leq \frac{2\alpha\gamma\epsilon}{(1-\gamma)^2} \tag{21}$$

$$\left| \eta_r(\pi_1) - \eta_r(\pi_2) - \sum_{t=0}^{\infty} \gamma^t \mathop{\mathbb{E}}_{s^{(t)} \sim \pi_2} \left[ \Delta \mathcal{A}(s^{(t)}) \right] \right| \leq \frac{2\alpha\gamma(2\alpha+1)\epsilon}{(1-\gamma)^2} \tag{22}$$

*Proof.* We first leverage Lemma 6 to derive Eq.23. Note that since $\pi_2$ is optimal under $r$, Remark 1 confirmed that $\mathcal{H}(\pi_2) = -\sum_{t=0}^{\infty} \gamma^t \mathop{\mathbb{E}}_{s^{(t)} \sim \pi_2} \left[ \mathop{\mathbb{E}}_{a^{(t)} \sim \pi_2} \left[ \mathcal{A}_{\pi_2}(s^{(t)}, a^{(t)}) \right] \right]$.

$$
\begin{aligned}
& \eta_r(\pi_1) - \eta_r(\pi_2) \\
= & \; (\eta_r(\pi_1) - \eta_r(\pi_2) - \mathcal{H}(\pi_2)) + \mathcal{H}(\pi_2) \\
= & \; \mathop{\mathbb{E}}_{\tau \sim \pi_1} \left[ \sum_{t=0}^{\infty} \gamma^t \mathcal{A}_{\pi_2}(s^{(t)}, a^{(t)}) \right] + \mathcal{H}(\pi_2) \\
= & \; \mathop{\mathbb{E}}_{\tau \sim \pi_1} \left[ \sum_{t=0}^{\infty} \gamma^t \mathcal{A}_{\pi_2}(s^{(t)}, a^{(t)}) \right] - \sum_{t=0}^{\infty} \gamma^t \mathop{\mathbb{E}}_{s^{(t)} \sim \pi_2} \left[ \mathop{\mathbb{E}}_{a^{(t)} \sim \pi_2} \left[ \mathcal{A}_{\pi_2}(s^{(t)}, a^{(t)}) \right] \right] \\
= & \; \sum_{t=0}^{\infty} \gamma^t \mathop{\mathbb{E}}_{s^{(t)} \sim \pi_1} \left[ \mathop{\mathbb{E}}_{a^{(t)} \sim \pi_1} \left[ \mathcal{A}_{\pi_2}(s^{(t)}, a^{(t)}) \right] - \mathop{\mathbb{E}}_{a^{(t)} \sim \pi_2} \left[ \mathcal{A}_{\pi_2}(s^{(t)}, a^{(t)}) \right] \right] + \\
& \; \sum_{t=0}^{\infty} \gamma^t \left( \mathop{\mathbb{E}}_{s^{(t)} \sim \pi_1} \left[ \mathop{\mathbb{E}}_{a^{(t)} \sim \pi_2} \left[ \mathcal{A}_{\pi_2}(s^{(t)}, a^{(t)}) \right] \right] - \mathop{\mathbb{E}}_{s^{(t)} \sim \pi_2} \left[ \mathop{\mathbb{E}}_{a^{(t)} \sim \pi_2} \left[ \mathcal{A}_{\pi_2}(s^{(t)}, a^{(t)}) \right] \right] \right) \\
= & \; \sum_{t=0}^{\infty} \gamma^t \mathop{\mathbb{E}}_{s^{(t)} \sim \pi_1} \left[ \Delta \mathcal{A}(s^{(t)}) \right] + \\
& \; \sum_{t=0}^{\infty} \gamma^t \left( \mathop{\mathbb{E}}_{s^{(t)} \sim \pi_1} \left[ \mathop{\mathbb{E}}_{a^{(t)} \sim \pi_2} \left[ \mathcal{A}_{\pi_2}(s^{(t)}, a^{(t)}) \right] \right] - \mathop{\mathbb{E}}_{s^{(t)} \sim \pi_2} \left[ \mathop{\mathbb{E}}_{a^{(t)} \sim \pi_2} \left[ \mathcal{A}_{\pi_2}(s^{(t)}, a^{(t)}) \right] \right] \right) \tag{23}
\end{aligned}
$$

We switch terms between Eq.23 and $\eta_r(\pi_1) - \eta_r(\pi_2)$, then use Lemma 9 to derive Eq.24.

$$
\begin{aligned}
& \left| \eta_r(\pi_1) - \eta_r(\pi_2) - \sum_{t=0}^{\infty} \gamma^t \mathop{\mathbb{E}}_{s^{(t)} \sim \pi_1} \left[ \Delta \mathcal{A}(s^{(t)}) \right] \right| \\
= & \; \left| \sum_{t=0}^{\infty} \gamma^t \left( \mathop{\mathbb{E}}_{s^{(t)} \sim \pi_1} \left[ \mathop{\mathbb{E}}_{a^{(t)} \sim \pi_2} \left[ \mathcal{A}_{\pi_2}(s^{(t)}, a^{(t)}) \right] \right] - \mathop{\mathbb{E}}_{s^{(t)} \sim \pi_2} \left[ \mathop{\mathbb{E}}_{a^{(t)} \sim \pi_2} \left[ \mathcal{A}_{\pi_2}(s^{(t)}, a^{(t)}) \right] \right] \right) \right| \\
\leq & \; \sum_{t=0}^{\infty} \gamma^t \cdot 2\max_{(s,a)} |\mathcal{A}_{\pi_2}(s, a)| \cdot (1 - (1-\alpha)^t) \leq \frac{2\alpha\gamma\max_{(s,a)}|\mathcal{A}_{\pi_2}(s, a)|}{(1-\gamma)^2} \tag{24}
\end{aligned}
$$

Alternatively, we can expand $\eta_r(\pi_2) - \eta_r(\pi_1)$ into Eq.25. During the process, $\mathcal{H}(\pi_2)$ is converted into $-\sum_{t=0}^{\infty}\gamma^t \mathop{\mathbb{E}}_{s^{(t)}\sim\pi_2}\left[\mathop{\mathbb{E}}_{a^{(t)}\sim\pi_2}\left[\mathcal{A}_{\pi_2}(s^{(t)},a^{(t)})\right]\right]$.

$$
\begin{aligned}
& \eta_r(\pi_1) - \eta_r(\pi_2) \\
=\ & (\eta_r(\pi_1) - \eta_r(\pi_2) - \mathcal{H}(\pi_2)) + \mathcal{H}(\pi_2) \\
=\ & \mathop{\mathbb{E}}_{\tau\sim\pi_1}\left[\sum_{t=0}^{\infty}\gamma^t \mathcal{A}_{\pi_2}(s^{(t)},a^{(t)})\right] + \mathcal{H}(\pi_2) \\
=\ & \sum_{t=0}^{\infty}\gamma^t \mathop{\mathbb{E}}_{s^{(t)}\sim\pi_1}\left[\mathop{\mathbb{E}}_{a^{(t)}\sim\pi_1}\left[\mathcal{A}_{\pi_2}(s^{(t)},a^{(t)})\right]\right] + \mathcal{H}(\pi_2) \\
=\ & \sum_{t=0}^{\infty}\gamma^t \mathop{\mathbb{E}}_{s^{(t)}\sim\pi_1}\left[\Delta A(s^{(t)}) + \mathop{\mathbb{E}}_{a^{(t)}\sim\pi_2}\left[\mathcal{A}_{\pi_2}(s^{(t)},a^{(t)})\right]\right] + \mathcal{H}(\pi_2) \\
=\ & \sum_{t=0}^{\infty}\gamma^t \mathop{\mathbb{E}}_{s^{(t)}\sim\pi_2}\left[\mathop{\mathbb{E}}_{a^{(t)}\sim\pi_1}\left[\mathcal{A}_{\pi_2}(s^{(t)},a^{(t)})\right] - \mathop{\mathbb{E}}_{a^{(t)}\sim\pi_2}\left[\mathcal{A}_{\pi_2}(s^{(t)},a^{(t)})\right] - \Delta\mathcal{A}(s^{(t)})\right] + \\
& \mathop{\mathbb{E}}_{s^{(t)}\sim\pi_1}\left[\Delta\mathcal{A}(s^{(t)}) + \mathop{\mathbb{E}}_{a^{(t)}\sim\pi_2}\left[\mathcal{A}_{\pi_2}(s^{(t)},a^{(t)})\right]\right] - \mathop{\mathbb{E}}_{\substack{s^{(t)}\sim\pi_2 \\ a^{(t)}\sim\pi_2}}\left[\mathcal{A}_{\pi_2}(s^{(t)},a^{(t)})\right] \\
=\ & \sum_{t=0}^{\infty}\gamma^t\left(\mathop{\mathbb{E}}_{s^{(t)}\sim\pi_1}\left[\mathop{\mathbb{E}}_{a^{(t)}\sim\pi_2}\left[\mathcal{A}_{\pi_2}(s^{(t)},a^{(t)})\right]\right] - 2\mathop{\mathbb{E}}_{s^{(t)}\sim\pi_2}\left[\mathop{\mathbb{E}}_{a^{(t)}\sim\pi_2}\left[\mathcal{A}_{\pi_2}(s^{(t)},a^{(t)})\right]\right]\right) + \\
& \sum_{t=0}^{\infty}\gamma^t\left(\mathop{\mathbb{E}}_{s^{(t)}\sim\pi_2}\left[\mathop{\mathbb{E}}_{a^{(t)}\sim\pi_1}\left[\mathcal{A}_{\pi_2}(s^{(t)},a^{(t)})\right]\right] - (\mathop{\mathbb{E}}_{s^{(t)}\sim\pi_2}\left[\Delta\mathcal{A}(s^{(t)})\right] - \mathop{\mathbb{E}}_{s^{(t)}\sim\pi_1}\left[\Delta\mathcal{A}(s^{(t)})\right])\right)
\end{aligned}
\tag{25}
$$

We switch terms between Eq.25 and $\eta_r(\pi_1) - \eta_r(\pi_2)$, then base on Lemma 8 and 9 to derive the inequality in Eq.26.

$$
\begin{aligned}
& \left|\eta_r(\pi_1) - \eta_r(\pi_2) - \sum_{t=0}^{\infty}\gamma^t \mathop{\mathbb{E}}_{s^{(t)}\sim\pi_2}\left[\Delta\mathcal{A}_{\pi}(s^{(t)},a^{(t)})\right]\right| \\
=\ & \Bigg|\eta_r(\pi_1) - \eta_r(\pi_2) - \\
& \qquad \sum_{t=0}^{\infty}\gamma^t\left(\mathop{\mathbb{E}}_{s^{(t)}\sim\pi_2}\left[\mathop{\mathbb{E}}_{a^{(t)}\sim\pi_1}\left[\mathcal{A}_{\pi_2}(s^{(t)},a^{(t)})\right]\right] - \mathop{\mathbb{E}}_{s^{(t)}\sim\pi_2}\left[\mathop{\mathbb{E}}_{a^{(t)}\sim\pi_2}\left[\mathcal{A}_{\pi_2}(s^{(t)},a^{(t)})\right]\right]\right)\Bigg| \\
=\ & \Bigg|\sum_{t=0}^{\infty}\gamma^t\left(\mathop{\mathbb{E}}_{s^{(t)}\sim\pi_2}\left[\Delta\mathcal{A}(s^{(t)})\right] - \mathop{\mathbb{E}}_{s^{(t)}\sim\pi_1}\left[\Delta\mathcal{A}(s^{(t)})\right]\right) - \\
& \qquad \sum_{t=0}^{\infty}\gamma^t\left(\mathop{\mathbb{E}}_{s^{(t)}\sim\pi_1}\left[\mathop{\mathbb{E}}_{a^{(t)}\sim\pi_2}\left[\mathcal{A}_{\pi_2}(s^{(t)},a^{(t)})\right]\right] - \mathop{\mathbb{E}}_{s^{(t)}\sim\pi_2}\left[\mathop{\mathbb{E}}_{a^{(t)}\sim\pi_2}\left[\mathcal{A}_{\pi_2}(s^{(t)},a^{(t)})\right]\right]\right)\Bigg| \\
\leq\ & \Bigg|\sum_{t=0}^{\infty}\gamma^t\left(\mathop{\mathbb{E}}_{s^{(t)}\sim\pi_2}\left[\Delta\mathcal{A}(s^{(t)})\right] - \mathop{\mathbb{E}}_{s^{(t)}\sim\pi_1}\left[\Delta\mathcal{A}(s^{(t)})\right]\right)\Bigg| + \\
& \qquad \Bigg|\sum_{t=0}^{\infty}\gamma^t\left(\mathop{\mathbb{E}}_{s^{(t)}\sim\pi_1}\left[\mathop{\mathbb{E}}_{a^{(t)}\sim\pi_2}\left[\mathcal{A}_{\pi_2}(s^{(t)},a^{(t)})\right]\right] - \mathop{\mathbb{E}}_{s^{(t)}\sim\pi_2}\left[\mathop{\mathbb{E}}_{a^{(t)}\sim\pi_2}\left[\mathcal{A}_{\pi_2}(s^{(t)},a^{(t)})\right]\right]\right)\Bigg| \\
\leq\ & \sum_{t=0}^{\infty}\gamma^t\left((1-(1-\alpha)^t)(4\alpha\max_{s,a}|\mathcal{A}_{\pi_2}(s,a)| + 2\max_{(s,a)}|\mathcal{A}_{\pi_2}(s,a)|)\right) \\
\leq\ & \frac{2\alpha\gamma(2\alpha+1)\max\limits_{s,a}|\mathcal{A}_{\pi_2}(s,a)|}{(1-\gamma)^2}
\end{aligned}
\tag{26}
$$

It is stated in Schulman et al. (2015) that $\max\limits_{s} D_{TV}(\pi_2(\cdot|s), \pi_1(\cdot|s)) \leq \alpha$. Hence, by letting $\alpha := \max\limits_{s} D_{TV}(\pi_2(\cdot|s), \pi_1(\cdot|s))$, Eq.23 and 26 still hold. Then, we have proved Theorem 4. $\qquad\square$

## B.2  OBJECTIVE FUNCTIONS OF REWARD OPTIMIZATION

To derive $J_{R,1}$ and $J_{R,2}$, we let $\pi_1 = \pi_P$ and $\pi_2 = \pi_A$. Then based on Eq.21 and 22 we derive the following upper-bounds of $\eta_r(\pi_P) - \eta_r(\pi_A)$.

$$\eta_r(\pi_P) - \eta_r(\pi_A) \leq \sum_{t=0}^{\infty} \gamma^t \mathop{\mathbb{E}}\limits_{s^{(t)} \sim \pi_P} \left[ \Delta\mathcal{A}(s^{(t)}) \right] + \frac{2\alpha\gamma(2\alpha+1)\epsilon}{(1-\gamma)^2} \tag{27}$$

$$\eta_r(\pi_P) - \eta_r(\pi_A) \geq \sum_{t=0}^{\infty} \gamma^t \mathop{\mathbb{E}}\limits_{s^{(t)} \sim \pi_A} \left[ \Delta\mathcal{A}(s^{(t)}) \right] - \frac{2\alpha\gamma\epsilon}{(1-\gamma)^2} \tag{28}$$

By our assumption that $\pi_A$ is optimal under $r$, we have $\mathcal{A}_{\pi_A} \equiv r$ Fu et al. (2018). This equivalence enables us to replace $\mathcal{A}_{\pi_A}$'s in $\Delta\mathcal{A}$ with $r$. As for the $\frac{2\alpha\gamma(2\alpha+1)\epsilon}{(1-\gamma)^2}$ and $\frac{2\alpha\gamma\epsilon}{(1-\gamma)^2}$ terms, since the objective is to maximize $\eta_r(\pi_A) - \eta_r(\pi_B)$, we heuristically estimate the $\epsilon$ in Eq.27 by using the samples from $\pi_P$ and the $\epsilon$ in Eq.28 by using the samples from $\pi_A$. As a result we have the objective functions defined as Eq.29 and 30 where $\delta_1(s,a) = \frac{\pi_P(a^{(t)}|s^{(t)})}{\pi_A(a^{(t)}|s^{(t)})}$ and $\delta_2 = \frac{\pi_A(a^{(t)}|s^{(t)})}{\pi_P(a^{(t)}|s^{(t)})}$ are the importance sampling probability ratio derived from the definition of $\Delta\mathcal{A}$; $C_1 \propto -\frac{\gamma\hat{\alpha}}{(1-\gamma)}$ and $C_2 \propto \frac{\gamma\hat{\alpha}}{(1-\gamma)}$ where $\hat{\alpha}$ is either an estimated maximal KL-divergence between $\pi_A$ and $\pi_B$ since $D_{KL} \geq D_{TV}^2$ according to Schulman et al. (2015), or an estimated maximal $D_{TV}^2$ depending on whether the reward function is Gaussian or Categorical. We also note that for finite horizon tasks, we compute the average rewards instead of the discounted accumulated rewards in Eq.30 and 29.

$$J_{R,1}(r; \pi_P, \pi_A) := \mathop{\mathbb{E}}\limits_{\tau \sim \pi_A} \left[ \sum_{t=0}^{\infty} \gamma^t \left( \delta_1(s^{(t)}, a^{(t)}) - 1 \right) \cdot r(s^{(t)}, a^{(t)}) \right] + C_1 \max\limits_{(s,a) \sim \pi_A} |r(s,a)| \tag{29}$$

$$J_{R,2}(r; \pi_P, \pi_A) := \mathop{\mathbb{E}}\limits_{\tau \sim \pi_P} \left[ \sum_{t=0}^{\infty} \gamma^t \left( 1 - \delta_2(s^{(t)}, a^{(t)}) \right) \cdot r(s^{(t)}, a^{(t)}) \right] + C_2 \max\limits_{(s,a) \sim \pi_P} |r(s,a)| \tag{30}$$

Beside $J_{R,1}, J_{R,2}$, we additionally use two more objective functions based on the derived bounds. W $J_{R,r}(r; \pi_A, \pi_P)$. By denoting the optimal policy under $r$ as $\pi^*$, $\alpha^* = \max\limits_{s \in \mathbb{S}} D_{TV}(\pi^*(\cdot|s), \pi_A(\cdot|s))$, $\epsilon^* = \max\limits_{(s,a^{(t)})} |\mathcal{A}_{\pi^*}(s, a^{(t)})|$, and $\Delta\mathcal{A}_A^*(s) = \mathop{\mathbb{E}}\limits_{a \sim \pi_A}[\mathcal{A}_{\pi^*}(s,a)] - \mathop{\mathbb{E}}\limits_{a \sim \pi^*}[\mathcal{A}_{\pi^*}(s,a)]$, we have the following.

$$
\begin{aligned}
& \eta_r(\pi_P) - \eta_r(\pi^*) \\
= \ & \eta_r(\pi_P) - \eta_r(\pi_A) + \eta_r(\pi_A) - \eta_r(\pi^*) \\
\leq \ & \eta_r(\pi_P) - \eta_r(\pi_A) + \sum_{t=0}^{\infty} \gamma^t \mathop{\mathbb{E}}\limits_{s^{(t)} \sim \pi_A} \left[ \Delta\mathcal{A}_A^*(s^{(t)}) \right] + \frac{2\alpha^*\gamma\epsilon^*}{(1-\gamma)^2} \\
= \ & \eta_r(\pi_P) - \sum_{t=0}^{\infty} \gamma^t \mathop{\mathbb{E}}\limits_{s^{(t)} \sim \pi_A} \left[ \mathop{\mathbb{E}}\limits_{a^{(t)} \sim \pi_A} \left[ r(s^{(t)}, a^{(t)}) \right] \right] + \\
& \quad \sum_{t=0}^{\infty} \gamma^t \mathop{\mathbb{E}}\limits_{s^{(t)} \sim \pi_A} \left[ \mathop{\mathbb{E}}\limits_{a^{(t)} \sim \pi_A} \left[ \mathcal{A}_{\pi^*}(s^{(t)}, a^{(t)}) \right] - \mathop{\mathbb{E}}\limits_{a^{(t)} \sim \pi^*} \left[ \mathcal{A}_{\pi^*}(s^{(t)}, a^{(t)}) \right] \right] + \frac{2\alpha^*\gamma\epsilon^*}{(1-\gamma)^2} \\
= \ & \eta_r(\pi_P) - \sum_{t=0}^{\infty} \gamma^t \mathop{\mathbb{E}}\limits_{s^{(t)} \sim \pi_A} \left[ \mathop{\mathbb{E}}\limits_{a^{(t)} \sim \pi^*} \left[ \mathcal{A}_{\pi^*}(s^{(t)}, a^{(t)}) \right] \right] + \frac{2\alpha^*\gamma\epsilon^*}{(1-\gamma)^2} \\
= \ & \mathop{\mathbb{E}}\limits_{\tau \sim \pi_P} \left[ \sum_{t=0}^{\infty} \gamma^t r(s^{(t)}, a^{(t)}) \right] - \mathop{\mathbb{E}}\limits_{\tau \sim \pi_A} \left[ \sum_{t=0}^{\infty} \gamma^t \frac{\exp(r(s^{(t)}, a^{(t)}))}{\pi_A(a^{(t)}|s^{(t)})} r(s^{(t)}, a^{(t)}) \right] + \frac{2\alpha^*\gamma\epsilon^*}{(1-\gamma)^2} \quad (31)
\end{aligned}
$$

Let $\delta_3 = \frac{\exp(r(s^{(t)}, a^{(t)}))}{\pi_A(a^{(t)}|s^{(t)})}$ be the importance sampling probability ratio. It is suggested in Schulman et al. (2017) that instead of directly optimizing the objective function Eq.31, optimizing a surrogate

objective function as in Eq.32, which is an upper-bound of Eq.31, with some small $\delta \in (0,1)$ can be much less expensive and still effective.

$$J_{R,3}(r; \pi_P, \pi_A) := \mathop{\mathbb{E}}_{\tau \sim \pi_P} \left[ \sum_{t=0}^{\infty} \gamma^t r(s^{(t)}, a^{(t)}) \right] -$$
$$\mathop{\mathbb{E}}_{\tau \sim \pi_A} \left[ \sum_{t=0}^{\infty} \gamma^t \min \left( \delta_3 \cdot r(s^{(t)}, a^{(t)}), clip(\delta_3, 1-\delta, 1+\delta) \cdot r(s^{(t)}, a^{(t)}) \right) \right] \tag{32}$$

Alternatively, we let $\Delta \mathcal{A}_P^*(s) = \mathop{\mathbb{E}}_{a \sim \pi_P} [\mathcal{A}_{\pi^*}(s,a)] - \mathop{\mathbb{E}}_{a \sim \pi^*} [\mathcal{A}_{\pi^*}(s,a)]$. The according to Eq.27, we have the following.

$$\eta_r(\pi_P) - \eta_r(\pi^*)$$
$$\leq \sum_{t=0}^{\infty} \gamma^t \mathop{\mathbb{E}}_{s^{(t)} \sim \pi_P} \left[ \Delta \mathcal{A}_P^*(s^{(t)}) \right] + \frac{2\alpha^* \gamma (2\alpha^* + 1)\epsilon^*}{(1-\gamma)^2}$$
$$= \sum_{t=0}^{\infty} \gamma^t \mathop{\mathbb{E}}_{s^{(t)} \sim \pi_P} \left[ \mathop{\mathbb{E}}_{a^{(t)} \sim \pi_P} \left[ \mathcal{A}_{\pi^*}(s^{(t)}, a^{(t)}) \right] - \mathop{\mathbb{E}}_{a^{(t)} \sim \pi^*} \left[ \mathcal{A}_{\pi^*}(s^{(t)}, a)^{(t)} \right] \right] + \frac{2\alpha^* \gamma (2\alpha^* + 1)\epsilon^*}{(1-\gamma)^2}$$
$$\tag{33}$$

Then a new objective function $J_{R,4}$ is formulated in Eq.34 where $\delta_4 = \frac{\exp(r(s^{(t)}, a^{(t)}))}{\pi_P(a^{(t)}|s^{(t)})}$.

$$J_{R,4}(r; \pi_P, \pi_A) := \mathop{\mathbb{E}}_{\tau \sim \pi_P} \left[ \sum_{t=0}^{\infty} \gamma^t r(s^{(t)}, a^{(t)}) \right] -$$
$$\mathop{\mathbb{E}}_{\tau \sim \pi_P} \left[ \sum_{t=0}^{\infty} \gamma^t \min \left( \delta_4 \cdot r(s^{(t)}, a^{(t)}), clip(\delta_4, 1-\delta, 1+\delta) \cdot r(s^{(t)}, a^{(t)}) \right) \right] \tag{34}$$

### B.3 INCORPORATING IRL ALGORITHMS

In our implementation, we combine PAGAR with GAIL and VAIL, respectively. When PAGAR is combined with GAIL, the meta-algorithm Algorithm 1 becomes Algorithm 2. When PAGAR is combined with VAIL, it becomes Algorithm 3. Both of the two algorithms are GAN-based IRL, indicating that both algorithms use Eq.1 as the IRL objective function. In our implementation, we use a neural network to approximate $D$, the discriminator in Eq.1. To get the reward function $r$, we follow Fu et al. (2018) and denote $r(s,a) = \log \left( \frac{\pi_A(a|s)}{D(s,a)} - \pi_A(a|s) \right)$ as mentioned in Section 1. Hence, the only difference between Algorithm 2 and Algorithm 1 is in the representation of the reward function. Regarding VAIL, since it additionally learns a representation for the state-action pairs, a bottleneck constraint $J_{IC}(D) \leq i_c$ is added where the bottleneck $J_{IC}$ is estimated from policy roll-outs. VAIL introduces a Lagrangian parameter $\beta$ to integrate $J_{IC}(D) - i_c$ in the objective function. As a result its objective function becomes $J_{IRL}(r) + \beta \cdot (J_{IC}(D) - i_c)$. VAIL not only learns the policy and the discriminator but also optimizes $\beta$. In our case, we utilize the samples from both protagonist and antagonist policies to optimize $\beta$ as in line 10, where we follow Peng et al. (2019) by using projected gradient descent with a step size $\delta$

In our implementation, depending on the difficulty of the benchmarks, we choose to maintain $\lambda$ as a constant or update $\lambda$ with the IRL loss $J_{IRL}(r)$ in most of the continuous control tasks. In *HalfCheetah-v2* and all the maze navigation tasks, we update $\lambda$ by introducing a hyperparameter $\mu$. As described in the maintext, we treat $\delta$ as the target IRL loss of $J_{IRL}(r)$, i.e., $J_{IRL}(r) \leq \delta$. In all the maze navigation tasks, we initialize $\lambda$ with some constant $\lambda_0$ and update $\lambda$ by $\lambda := \lambda \cdot \exp(\mu \cdot (J_{IRL}(r) - \delta))$ after every iteration. In *HalfCheetah-v2*, we update $\lambda$ by $\lambda := max(\lambda_0, \lambda \cdot \exp(\mu \cdot (J_{IRL}(r) - \delta)))$ to avoid $\lambda$ being too small. Besides, we use PPO Schulman et al. (2017) to train all policies in Algorithm 2 and 3.

## C EXPERIMENT DETAILS

This section presents some details of the experiments and additional results.

**Algorithm 2** GAIL w/ PAGAR

**Input**: Expert demonstration $E$, discriminator loss bound $\delta$, initial protagonist policy $\pi_P$, antagonist policy $\pi_A$, discriminator $D$ (representing $r(s,a) = \log\left(\frac{\pi_A(a|s)}{D(s,a)} - \pi_A(a|s)\right)$), Lagrangian parameter $\lambda$, iteration number $i = 0$, maximum iteration number $N$

**Output**: $\pi_P$

1: **while** iteration number $i < N$ **do**
2:     Sample trajectory sets $\mathbb{D}_A \sim \pi_A$ and $\mathbb{D}_P \sim \pi_P$
3:     Estimate $J_{RL}(\pi_A; r)$ with $\mathbb{D}_A$
4:     Optimize $\pi_A$ to maximize $J_{RL}(\pi_A; r)$.
5:     Estimate $J_{RL}(\pi_P; r)$ with $\mathbb{D}_P$; $J_{PPO}(\pi_P; \pi_A, r)$ with $\mathbb{D}_P$ and $\mathbb{D}_A$;
6:     Optimize $\pi_P$ to maximize $J_{RL}(\pi_P; r) + J_{PPO}(\pi_P; \pi_A, r)$.
7:     Estimate $J_{PAGAR}(r; \pi_P, \pi_A)$ with $\mathbb{D}_P$ and $\mathbb{D}_A$
8:     Estimate $J_{IRL}(\pi_A; r)$ with $\mathbb{D}_A$ and $E$ by following the IRL algorithm
9:     Optimize $D$ to minimize $J_{PAGAR}(r; \pi_P, \pi_A) + \lambda \cdot max(J_{IRL}(r) + \delta, 0)$
10: **end while**
11: **return** $\pi_P$

---

**Algorithm 3** VAIL w/ PAGAR

**Input**: Expert demonstration $E$, discriminator loss bound $\delta$, initial protagonist policy $\pi_P$, antagonist policy $\pi_A$, discriminator $D$ (representing $r(s,a) = \log\left(\frac{\pi_A(a|s)}{D(s,a)} - \pi_A(a|s)\right)$), Lagrangian parameter $\lambda$ for PAGAR, iteration number $i = 0$, maximum iteration number $N$, Lagrangian parameter $\beta$ for bottleneck constraint, bounds on the bottleneck penalty $i_c$, learning rate $\mu$.

**Output**: $\pi_P$

1: **while** iteration number $i < N$ **do**
2:     Sample trajectory sets $\mathbb{D}_A \sim \pi_A$ and $\mathbb{D}_P \sim \pi_P$
3:     Estimate $J_{RL}(\pi_A; r)$ with $\mathbb{D}_A$
4:     Optimize $\pi_A$ to maximize $J_{RL}(\pi_A; r)$.
5:     Estimate $J_{RL}(\pi_P; r)$ with $\mathbb{D}_P$; $J_{PPO}(\pi_P; \pi_A, r)$ with $\mathbb{D}_P$ and $\mathbb{D}_A$;
6:     Optimize $\pi_P$ to maximize $J_{RL}(\pi_P; r) + J_{PPO}(\pi_P; \pi_A, r)$.
7:     Estimate $J_{PAGAR}(r; \pi_P, \pi_A)$ with $\mathbb{D}_P$ and $\mathbb{D}_A$
8:     Estimate $J_{IRL}(\pi_A; r)$ with $\mathbb{D}_A$ and $E$ by following the IRL algorithm
9:     Estimate $J_{IC}(D)$ with $\mathbb{D}_A, \mathbb{D}_P$ and $E$
10:     Optimize $D$ to minimize $J_{PAGAR}(r; \pi_P, \pi_A) + \lambda \cdot max(J_{IRL}(r) + \delta, 0) + \beta \cdot J_{IC}(D)$
11:     Update $\beta := \max\left(0, \beta - \mu \cdot (\frac{J_{IC}(D)}{3} - i_c)\right)$
12: **end while**
13: **return** $\pi_P$

---

## C.1 EXPERIMENTAL DETAILS

**Network Architectures**. Our algorithm involves a protagonist policy $\pi_P$, and an antagonist policy $\pi_A$. In our implementation, the two policies have the same structures. Each structure contains two neural networks, an actor network, and a critic network. When associated with GAN-based IRL, we use a discriminator $D$ to represent the reward function as mentioned in Appendix B.3.

- **Protagonist and Antagonist policies**. We prepare two versions of actor-critic networks, a fully connected network (FCN) version, and a CNN version, respectively, for the Mujoco and Mini-Grid benchmarks. The FCN version, the actor and critic networks have 3 layers. Each hidden layer has 100 neurons and a $tanh$ activation function. The output layer output the mean and standard deviation of the actions. In the CNN version, the actor and critic networks share 3 convolutional layers, each having 5, 2, 2 filters, $2 \times 2$ kernel size, and $ReLU$ activation function. Then 2 FCNs are used to simulate the actor and critic networks. The FCNs have one hidden layer, of which the sizes are 64.

- **Discriminator $D$ for PAGAR-based GAIL in Algorithm 2**. We prepare two versions of discriminator networks, an FCN version and a CNN version, respectively, for the Mujoco

and Mini-Grid benchmarks. The FCN version has 3 linear layers. Each hidden layer has 100 neurons and a $tanh$ activation function. The output layer uses the $Sigmoid$ function to output the confidence. In the CNN version, the actor and critic networks share 3 convolutional layers, each having 5, 2, 2 filters, $2 \times 2$ kernel size, and $ReLU$ activation function. The last convolutional layer is concatenated with an FCN with one hidden layer with 64 neurons and $tanh$ activation function. The output layer uses the $Sigmoid$ function as the activation function.

- **Discriminator** $D$ **for PAGAR-based VAIL in Algorithm 3**. We prepare two versions of discriminator networks, an FCN version and a CNN version, respectively, for the Mujoco and Mini-Grid benchmarks. The FCN version uses 3 linear layers to generate the mean and standard deviation of the embedding of the input. Then a two-layer FCN takes a sampled embedding vector as input and outputs the confidence. The hidden layer in this FCN has 100 neurons and a $tanh$ activation function. The output layer uses the $Sigmoid$ function to output the confidence. In the CNN version, the actor and critic networks share 3 convolutional layers, each having 5, 2, 2 filters, $2 \times 2$ kernel size, and $ReLU$ activation function. The last convolutional layer is concatenated with a two-layer FCN. The hidden layer has 64 neurons and uses $tanh$ as the activation function. The output layer uses the $Sigmoid$ function as the activation function.

**Hyperparameters** The hyperparameters that appear in Algorithm 3 and 3 are summarized in Table 2 where we use N/A to indicate using the maximal $\delta$ as mentioned in Section 4.2, in which case we let $\mu = 0$. Otherwise, the values of $\mu$ and $\delta$ vary depending on the task and IRL algorithm. The parameter $\lambda_0$ is the initial value of $\lambda$ as explained in Appendix B.3.

| Parameter | Continuous Control Domain | Partially Observable Domain |
|---|---|---|
| Policy training batch size | 64 | 256 |
| Discount factor | 0.99 | 0.99 |
| GAE parameter | 0.95 | 0.95 |
| PPO clipping parameter | 0.2 | 0.2 |
| $\lambda_0$ | 1e3 | 1e3 |
| $\sigma$ | 0.2 | 0.2 |
| $i_c$ | 0.5 | 0.5 |
| $\beta$ | 0.0 | 0.0 |
| $\mu$ | VAIL(HalfCheetah): 0.5; others: 0.0 | VAIL: 1.0; GAIL: 1.0 |
| $\delta$ | VAIL(HalfCheetah): 1.0; others: N/A | VAIL: 0.8; GAIL: 1.2 |

Table 2: Hyperparameters used in the training processes

**Expert Demonstrations.** Our expert demonstrations all achieve high rewards in the task. The number of trajectories and the average trajectory total rewards are listed in Table 3.

## C.2 ADDITIONAL RESULTS

We append the results in three Mujoco benchmarks: *Hopper-v2*, *InvertedPendulum-v2* and *Swimmer-v2* in Figure 7. Algorithm 1 performs similarly to VAIL and GAIL in those two benchmarks. IQ-learn does not perform well in Walker2d-v2 but performs better than ours and other baselines by a large margin.

| Task | Number of Trajectories | Average Tot.Rewards |
|---|---|---|
| Walker2d-v2 | 10 | 4133 |
| HalfCheetah-v2 | 100 | 1798 |
| Hopper-v2 | 100 | 3586 |
| InvertedPendulum-v2 | 10 | 1000 |
| Swimmer-v2 | 10 | 122 |
| DoorKey-6x6-v0 | 10 | 0.92 |
| SimpleCrossingS9N1-v0 | 10 | 0.93 |

Table 3: The number of demonstrated trajectories and the average trajectory rewards

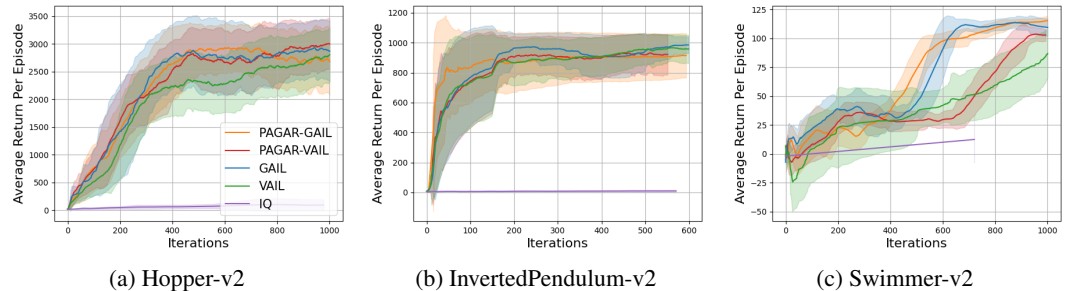

(a) Hopper-v2      (b) InvertedPendulum-v2      (c) Swimmer-v2

Figure 7: Comparing Algorithm 1 with baselines. The suffix after each 'PAGAR-' indicates which IRL algorithm is utilized in Algorithm 1. The $y$ axis is the average return per step. The $x$ axis is the number of iterations in GAIL, VAIL, and ours. The policy is executed between each iteration for $2048$ timesteps for sample collection. One exception is that IQ-learn updates the policy at every timestep, making its actual number of iterations $2048$ times larger than indicated in the figures.

### C.3 INFLUENCE OF REWARD HYPOTHESIS SPACE

In addition to the *DoorKey-6x6-v0* environment, we also tested PAGAR-GAIL and GAIL in *SimpleCrossingS9N2-v0* environment. The results are shown in Figure 8.

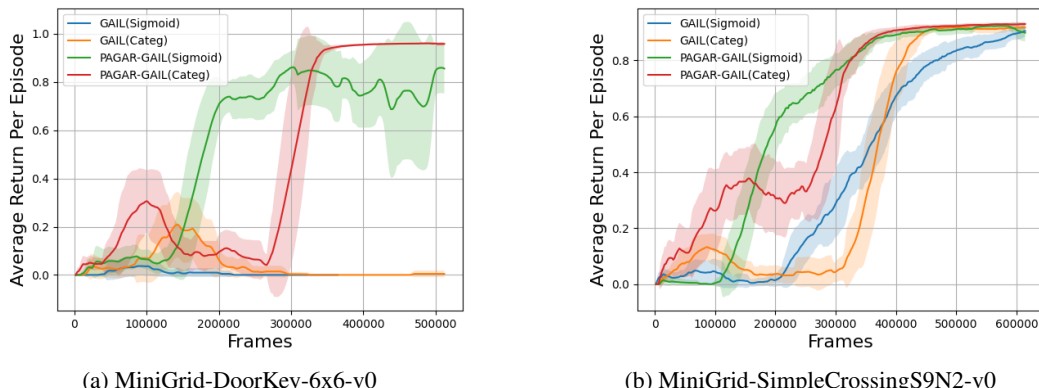

(a) MiniGrid-DoorKey-6x6-v0      (b) MiniGrid-SimpleCrossingS9N2-v0

Figure 8: Comparing Algorithm 1 with baselines. The prefix 'protagonist_GAIL' indicates that the IRL algorithm utilized in Algorithm 1 is the same as in GAIL. The '_Sigmoid' and '_Categ' suffixes indicate whether the output layer of the discriminator is using the $Sigmoid$ function or Categorical distribution. The $x$ axis is the number of sampled frames. The $y$ axis is the average return per episode.

