# OpenReview forum: "PAGAR: Taming Reward Misalignment in Inverse Reinforcement Learning-Based Imitation Learning with Protagonist Antagonist Guided Adversarial Reward"
_ICLR.cc/2024/Conference — Submitted to ICLR 2024_

### Official Review · Reviewer_58y7 · 2023-10-31

**Soundness:** 3 good
**Presentation:** 3 good
**Contribution:** 3 good
**Rating:** 6
**Confidence:** 3

**Summary:**

This paper addresses the reward misalignment problem in IRL, which refers to the problem that reward maximization does not guarantee task success. To address this problem, PAGAR introduces both an antagonist policy - an optimal policy trained to perform similarly to the expert - and a protagonist policy that is trained to be closer to the antagonist policy using a PPO-style objective function. PAGAR then derives a reward function that maximizes the performance difference between the protagonist policy and the antagonist policy. When the performance margin remains within a certain bound, the protagonist policy's performance with the updated reward function does not fall below a certain threshold, indicating reward alignment. By training the protagonist policy in a minimax manner, the authors argue that an aligned reward function can be identified and used to optimize the policy. The authors demonstrate PAGAR's performance in partially observable navigation tasks and transfer environments.

**Strengths:**

The authors conducted a thorough analysis of reward function alignment and successfully applied their insights within the IRL framework. They demonstrated that the PAGAR-(IRL) algorithms outperform baselines, requiring fewer samples and enabling zero-shot imitation learning in transfer environments.

**Weaknesses:**

There are a few minor typos present:

- On the 2nd line from the bottom in Figure 1, $U_{r^-}(\pi^-) \rightarrow U_{r^-}(\pi^*)$.
- On the 5th line from the bottom, please confirm whether the direction of the inequality sign is correct.
- On the first line in section 2, Related works, contextx should be changed to contexts.

**Questions:**

I'm curious if the antagonist policy can be a non-optimal policy. In Figure 2 (b), VAIL does not appear to have enough successful episodes to qualify as an optimal policy and serve as a proxy. Would it be acceptable for the policy to have non-zero returns instead of being fully optimal?

---

> ### Author Response · Authors · 2023-11-16
>
> ## 5th line from the bottom ... inequality sign
>
> The sign is correct, when $U_{r^-}(E)-max_\pi U_{r^-}(\pi) > U_{r^+}(E) - max_\pi U_{r^+}(\pi)$, reward function $r^-$ will be more 'optimal' than $r^+$.
> We use this inequality to show that we must consider the case where a sub-optimal reward function can align with the task while the optimal reward function is misaligned.
>
>
> ## ... non-optimal policy ...
>
> In theory, the antagonist policy has to be optimal, while in practice, we have to train the antagonist policy together with the protagonist policy from scratch.
> The fact that VAIL alone not being able to produce successful episodes does not directly relate to whether running VAIL in the PAGAR framework can produce better results.
> This is because the reward function in the PAGAR framework is optimized to increase the protagonist antagonist induced regret in Eq (2). This optimization incentivizes searching for a reward function that can makes the antagonist policy outperforms the protagonist policy.
> Given that inverse RL can have multiple solutions -- multiple reward functions aligned with the demonstrations, PAGAR-based IL avoids those reward functions that are difficult for any policy to achieve a high utility, since this will result in both the antagonist and the protagonist policy having low performance.

---

### Official Review · Reviewer_rmTC · 2023-10-31

**Soundness:** 2 fair
**Presentation:** 1 poor
**Contribution:** 3 good
**Rating:** 5
**Confidence:** 3

**Summary:**

The paper is inspired by unsupervised environment design (UED) [Dennis et al. 2020] which optimizes a protagonist policy an antagonist policy and the dynamics parameters with respect to the regret (performance difference between antagonist and protagonist; protagonist aims to minimize the regret, antagonist and "environment" aim to maximize regret). However, in contrast to that prior work, the current submission tackles the imitation learning setting, by maximizing the regret with respect to a reward function instead of dynamics parameter. The motivation for this approach is to learn policies that perform well across a distribution of reward function making them less susceptible to reward misalignment, where a policy performs well on an inferred reward function but still fails in the underlying task.

**Strengths:**

Although there is a close connection to UED, the application to imitation learning is quite original.

I also think that the problem setting, focusing on solving the actual underlying task, is very relevant.

**Weaknesses:**

1. Clarity
------------
I'm struggling to make sense of some parts of the work. For now, I view these as clarity issues that I hope the authors can address in the rebuttal, but there may also be issues regarding soundness or relevance:

- I'm not fully convinced of the motivation for the method. The paper argues that existing IL/IRL methods may fail in optimizing the underlying task because they learn a policy that optimizes a misaligned reward function. However, I do not think that the typically used problem formulation suffer from such problems. For example IL methods are often formulated as divergence minimization problems, and assuming that they succeed in bringing the divergence to zero, they would perform as well as the expert on any reward function. Of course, in practice we will typically not achieve zero divergence due function approximations, limited data and potentially unstable optimization, but I don't think that these challenges are specific to the problem formulation and they affect the proposed method in similar way. For some IRL methods, it was even shown that the optimal policy for the learned reward function may outperform the (suboptimal) expert, in the sense that it performs no worse but potentially better on every reward function within the hypothesis space. With that in mind, I also find Example 1 quite misleading. The example argues that IRL would lead to a bad policy in a simple toy problems, by arguing that a constant reward function would solve the IRL problem. However, already Ng & Russell [2000] noted, that such IRL formulation would be illposed, and additional constraints / objectives are required. For example, MaxEnt-IRL would maximize the likelihood for a MaxEnt-policy, where a constant reward function (leading to a uniform policy) would certainly not be a solution in the toy problem.

- The conditions used in Theorem 1 are not satisfiable if the hypothesize space of the reward function includes a reward function for which there is no gap between the best performing failing policy and a succeeding policy. For example, if a task is considered to succeed when the distance to a goal is below a certain threshold, and the reward uses a quadratic penalty on the distance to the threshold as cost, we can get arbitrary close in terms of reward to a succeeding policy while still failing. Potentially related to this, it not clear to me how the method optimizes with respect to a distribution $\mathcal{P}\_{\pi}(r)$ over reward functions. Where does this distribution appear in Algorithm 1?

- Section 4.2 defines a set $R\_{E,\delta} =  \\{ r \| U\_{r}(E) - \underset{\pi \in \Pi}{\max} U\_{r}(\pi) \ge \delta \\}$ where $\delta$ is chosen to be strictly positive. Wouldnt this set be empty if the expert policy is in $\Pi$?

- I am wondering how the success of a policy or its performance can be deterministic, given that both policy and environment are stochastic.

- The transfer experiment is not clear to me. Is the learned policy transferred to the new environment, or is it learned from scratch in the new environment, using demonstration from the source environment?

2. Evaluation
-------------
The experimental results in the continuous domain are not convincing. For example, IQ-Learn should clearly outperform GAIL in these environments. The paper states that it should perform better if a bigger replay buffer was used, but then I wonder why no adequate hyperparameter tuning was performed for the comparisons,

**Questions:**

For questions, please refer to my comments under "Weaknesses".

---

> ### Author Response · Authors · 2023-11-16
>
> > ## ... maximize the likelihood for a MaxEnt-policy ...
>
> Maximum entropy is a heuristic used to reduce reward ambiguity and not reward misalignment.
> In fact, MaxEnt cannot eliminate reward ambiguity even if we have infinite demonstrations ([Skalse. et al., 2022], which we cite in the first paragraph of Introduction).
>
> Reward ambiguity and reward-task misalignment are two different concepts.
> Reward ambiguity is the problem of many different reward functions in the reward hypothesis space can explain the expert demonstrations equally well.
> On the other hand, reward-task misalignment
> is the problem of policies that achieve a high utility under a reward function failing to finish the task (the well-known notion of reward hacking is a manifestation of this [Pan. et al., 2022], which we cite at the end of first paragraph).
>
> Maximum entropy likelihood may lead to a policy that is highly probable in reproducing the demonstrated trajectories.
> It is equivalent to assuming that the underlying task is simply to generate the demonstrated trajectories with the highest probability.
> Our work does not make such an assumption. As shown by our definition of $R_{E,\delta}$, we include not only the optimal but also the sub-optimal solutions of inverse RL as candidate reward functions.
>
> > ## ... The conditions used in Theorem 1 are not satisfiable ... #
>
> Firstly, in Definition 1, the intervals $S_r$ and $F_r$ are defined such that a policy's utility falling within $S_r$ or $F_r$ is deemed a sufficient but not necessary condition for determining success or failure. Put differently, if a policy $\pi$ achieves a utility in $S_r$, it is guaranteed to be successful, and if $\pi$ achieves a utility in $F_r$, it is guaranteed to be a failure. Note that $S_r$ and $F_r$ are not required to encompass all successful or failing policies' utilities. These intervals may also be very small for certain task. In the extreme scenario, $S_r$ could be a singleton, representing the maximum achievable policy utility under $r$, if the optimal policy under $r$ can successfully complete the task. Meanwhile, $F_r$ can be empty.
> The impact is that if all reward functions in the set $R$ have very small $S_r$ and $F_r$ intervals, $MinimaxRegret$ may have to achieve a low regret to produce a successful policy.
>
> Also, the conditions about the width of S and F intervals are sufficient conditions for $MinimaxRegret$ to learn a successful policy. Suppose all the reward functions in the candidate reward function set are task-aligned. Then, even if they all match the description
>  mentioned by the reviewer, i.e., `no gap between  $S_r$ and $F_r$`, $MinimaxRegret$ can still induce a successful policy in the task.
> > Proof
>    * There must be a policy $\pi^+$ that makes $Regret(\pi^+, r)\leq|S_r|$ hold for all reward function $r$'s. Because its utility cannot be in the S interval under a reward function and in the F interval under another.
>    * For any policy $\pi^-$ that fails the task, $Regret(\pi^+, r)>|S_r|$ holds for all reward function $r$'s. Because if there exists an $r$ under which $Regret(\pi^-, r) \leq  |S_r|$, $\pi^-$ is guaranteed to succeed in the task, contradicting the assumption.
>    * $MinimaxRegret$ will pick $\pi^+$ over $\pi^-$ as its solution because $Regret(\pi^+, r)<Regret(\pi^-, r)$ under all $r$'s
>
> > ## ... Potentially related to this, it not clear to me how the
>
> Table 1 shows an equivalent objective function of $MinimaxRegret$
> It only describes the analytical solution of PAGAR.
> Our approach to PAGAR is not based on Table 1.
>
> > ## ... $\delta$  is chosen to be strictly positive ...
>
> We mentioned after Corollary 2 that $\delta$ cannot be larger than the negative minimal loss of inverse RL.
> If the minimal loss of inverse RL is non-negative, i.e., the expert policy is in the policy set, $\delta$ will be non-positive.
>
> > ## ...the success of a policy or its performance can be deterministic ...
>
> The 'success' here indicates the ability of the policy to finish the task, not whether the policy finishes the task in a single run.
> For instance, a success criterion can be "the probability of reaching the state must be higher than 'p'".
> If a reward function is well-designed, the success of this policy can be guaranteed if the policy's expected return exceeds a certain threshold.
>
>
> > ## ...  IQ-Learn should clearly outperform ...
>
> In our continuous control experiments, the x-axis represents policy update iterations, not policy roll-out steps.
> IQ-learn's implementation is based on soft-actor-critic which is an off-policy algorithm that conducts several policy updates between consecutive interactions with the environment.
> The size of the replay buffer allocated for IQ-learn is consistent with the sample size used for PPO which is the RL algorithm used in our implementation.
>
> > ## ... transfer experiment
>
> The demonstrations are sampled in one environment, Algorithm 1 (reward learning, policy learning) is conducted in another environment.

---

> > ### Comment · Reviewer_rmTC · 2023-11-23
> > **Concerns have not been adequately addressed**
> >
> > > Maximum entropy is a heuristic used to reduce reward ambiguity and not reward misalignment. In fact, MaxEnt cannot eliminate > reward ambiguity even if we have infinite demonstrations ([Skalse. et al., 2022], which we cite in the first paragraph of Introduction).
> >
> > I would not reduce the principle of Maximum Entropy to a mere heuristic. MaxEnt-IRL chooses among all models of the expert the one that makes least assumption, which is arguably the most principled choice! Please also note that MaxEnt-IRL is well-posed. The dual function is convex, which implies that there is a single reward that solves the MaxEnt-IRL optimization problem. If we are talking about more modern approaches that maximize the likelihood of MaxEnt-RL policy, but optimize over a NN reward function instead of a linear reward function, then I agree that there is reward ambiguity since potential-based reward shaping does not affect the (MaxEnt)-optimal policy. However, even then, the different reward functions all result in the same behavior on the given environment, and hence, reward ambiguity only becomes an issues when we want to apply the learned reward function to MDPs with different dynamics.
> >
> > I do not see how the concept of reward misalignment can apply to IRL. If the agent reproduces the behavior of the expert and we assume that the expert solves the task, the agent will also solve the task. Typical IRL formulation, will find a reward function that when optimized either matches the expert exactly, or derivates if this provably does not diminish reward on all reward functions in the hypothesis space.
> >
> > >Maximum entropy likelihood may lead to a policy that is highly probable in reproducing the demonstrated trajectories. It is >equivalent to assuming that the underlying task is simply to generate the demonstrated trajectories with the highest probability.
> >
> > This claim is not substantiated and wrong. The underlying task would always be to maximize the reward function. Maximizing the MaxEnt reward function would not necessarily result in the demonstrated trajectories to obtain highest probability. Even when MaxEnt-optimizing the learned reward function, the maximum likelihood trajectory generated of the policy might in general not coincide with any demonstration.
> >
> > Also, my comments regarding Example 1 have not been addressed!
> >
> > > We mentioned after Corollary 2 that $\delta$ cannot be larger than the negative minimal loss of inverse RL. If the minimal loss of inverse RL is non-negative, i.e., the expert policy is in the policy set, $\delta$ will be non-positive.
> >
> > The text after Corollary 2 mentions that $R\_{E,\delta}$ is learned for a given $\delta$. Table 2, also lists $\delta$ as a hyperparameter. These hyperparamters are non-sensical, at least if we assume the expert policy to be in the hypothesis space.
> >
> > > In our continuous control experiments, the x-axis represents policy update iterations, not policy roll-out steps. IQ-learn's implementation is based on soft-actor-critic which is an off-policy algorithm that conducts several policy updates between consecutive interactions with the environment. The size of the replay buffer allocated for IQ-learn is consistent with the sample size used for PPO which is the RL algorithm used in our implementation.
> >
> > This does not address my complain that the evaluation does not show a fair comparison. For a fair comparison, the hyperparameters should be chosen suitable to the method. Is also not fair to compare SAC based method with PPO based method based with respect to the number of policy updates, since SAC usually updates the policy after a single transition. Instead the comparison should be with respect to the number of transitions. For evaluating the computational overhead of the single-step updates, additionally the computational time could be shown.
> >
> > Plotting over the number of policy updates is severely misguiding the reader to overestimate the relative performance between the proposed method and IQ-Learn!
> >
> > I will reduced my score from 5 to 3.

---

> ### Author Response · Authors · 2023-11-23
>
> >  If the agent reproduces the behavior of the expert and we assume that the expert solves the task, the agent will also solve the task.
>
> A policy being able to reproduce the demonstrated trajectories does not imply the policy is ready for the task. In a stochastic environment, any stochastic policy can reproduce the demonstrated trajectories ... We have reiterated that the outcomes of individual attempts or instances do not solely determine success. If by `reproduces the behavior` you mean to recover the expert policy, the problem is that IRL can fail to recover expert policy for so many reasons ... We have cited the literature in the paper and in our rebuttal.
>
> > This claim is not substantiated and wrong ...
>
> Our claim is substantiated and correct. We recommend revisiting `Ziebart, Brian D., et al. Maximum entropy inverse reinforcement learning` and searching for the keyword 'likelihood'. It is beyond doubt that max-ent IRL determines the reward weight by maximizing the likelihood of the max-ent policy generating the demonstrated trajectories. Max-Ent principle is not the only factor at play.
>
> > The underlying task would always be to maximize the reward function.
>
> This does not contradict our argument. The policy is trained with a reward function that is learned based on a maximum likelihood objective function.
>
> > ... at least if we assume the expert policy to be in the hypothesis space
>
> Our paper never made such an assumption at all.
> It is common knowledge that IRL can be affected by various factors, including a limited number of expert demonstrations, expert policy not being in the hypothesis space, learning reward function in MDPs with different dynamics ...
> And our method is to mitigate the reward misalignment issue in challenging and varied conditions.
> While Theorem 3 implicitly covers the ideal condition where IRL can reach Nash Equilibrium, all other theories do not account for the ideal condition.
> Our experiments in MiniGrid aim to exhibit PAGAR's ability when those factors are present.

---

### Official Review · Reviewer_BsED · 2023-11-01

**Soundness:** 3 good
**Presentation:** 1 poor
**Contribution:** 2 fair
**Rating:** 3
**Confidence:** 3

**Summary:**

The paper introduces a method for reducing reward misalignment in RL and IL settings. The method follows the paradigm introduced in PAIRED, where there is a triple-level objective: the protagonist tries to minimize regret, a design agent selects a reward function to maximize regret, and the antagonist also tries to maximize regret. The paper first formalizes task-reward alignment using the introduction of success and failure intervals. Regret is defined as the difference in expected return achieved by the best-case antagonist and the protagonist. Then the authors provide a theoretical result demonstrating that, under certain conditions, the protagonist policy optimizing for minimax regret can guarantee avoiding failing the task and possibly succeed as well. Next the authors analyze the IL setting, where expert demos are provided in lieu of a reward function. Then they provide an algorithm for PAGAR-based IL, which alternates between the policy optimization steps (for P and A) and reward selection step. The policy optimization step is fairly straightforward using a PPO-style objective. Reward selection for regret maximization uses bounds on the difference in expected return between P and A using only the samples of one policy. Experiments are conducted in maze navigation and MuJoCo continuous control tasks, both demonstrating PAGAR can mitigate reward misalignment.

**Strengths:**

The strengths of this paper are 1) the novelty in applying PAIRED's paradigm to reward misalignment and 2) the very appreciable amount of technical novelty. Regarding the latter, the authors formalize task-reward misalignment (which is generally only explained intuitively), define regret for the reward misalignment mitigation problem, and give algorithms for both RL and IL with PAGAR. The authors provide theoretical results showing that under certain conditions, PAGAR's minimax regret objective, if optimized perfectly, is able to guarantee avoiding task failure and possibly succeed, in both the RL and IL settings. These are important and strong results.

Experiments are provided in both gridworlds/discrete envs and continuous envs.

**Weaknesses:**

The paper was very difficult to follow. There was quite a lot of new notation introduced for a fairly intuitive concept, and it was prohibitively difficult to keep track of all the new notation, definitions, and theorems. I was not able to fully appreciate the implications of the theory. I would be willing to raise my score if the paper were made significantly more readable, since this is an interesting and novel idea.

In particular, the paper does not answer the question of how the set R (under which reward optimization is performed) is chosen in the RL setting, instead redirecting to the IL setting. Is PAGAR meant to be applied in the RL setting or is the selection of R prohibitively difficult and therefore PAGAR as described is only applicable in practice to IL? It would help to make this more explicit.

In the IL setting, it seems that generation of the R_E,delta set is difficult, but it is not discussed how to obtain such a set. Example 1 does not provide intuition for the basis functions it chose to parameterize R_E so it was difficult to understand. Is the discussion at the bottom of pg 5 meant to be a construction of R_E or just a non-constructive description?

The paper is missing a discussion of the computational complexity of PAGAR, both theoretically and empirically. This is a crucial property of the method.

Experiments are only conducted with a very limited number of demonstrations (10). I don't find this realistic for real-world deployment where we care about alignment. Is this because providing more demonstrations would reduce misalignment and thus the need for a complicated method like PAGAR?

**Questions:**

- Is PAGAR meant to be applied in the RL setting or is the selection of R prohibitively difficult and therefore PAGAR as described is only applicable in practice to IL?
- Is the discussion at the bottom of pg 5 meant to be a construction of R_E or just a non-constructive description?
- how difficult is it to specify success/failure for non-binary-outcome tasks in general?
- definition 2 is not a definition as written, it is a claim about the method.
- how strong of a condition is (2) in theorem 1?
- some of the lines in figure 2 look like they are still increasing and not yet converged. can you run them for longer?
- in figure 2 does timesteps count just the protagonist or everyone? if not, can you compare with # FLOPS as well? I imagine PAGAR is quite a bit more expensive.
- is AIRL a suitable baseline? it's also disentangled from transition dynamics which is related to your objective for the zero-shot transfer experiments.
- can you provide more details what exactly the transfer is? is it transferring from one random instantiation of a family of envs to another?
- It is odd to put a definition in the intro, I would suggest keeping the intro at a high level and moving the definition to a following section.
- I noticed that often a definition would introduce many new objects at once and require reading an equation given later in order to appreciate a previous sentence. I would recommend re-ordering the presentation of material such that the flow of understanding is always moving forward, which can be accomplished by introducing new terms/definitions/concepts one step at a time and building up from the first object the reader should know (and can understand without needing to pull in other objects first).

---

> ### Author Response · Authors · 2023-11-16
>
> > ## does not answer the question of how the set $R$ is chosen in the RL setting ... Is PAGAR meant to be applied in the RL setting ... Only applicable in practice to IL
>
> We would like to clarify that
> Section 4.1, "RL with PAGAR", is about examining the conditions for $R$ under which $MinimaxRegret(R)$ can generate a policy to avoid task failure or guarantee task success.
> It does not deal with how to construct such an $R$ in an RL training scenario.
> It is meant to support the main objective of the paper, which is mitigating the reward alignment problem in IRL-based IL, since IRL-based IL involves inferring a reward from the demonstrations and performing RL with the inferred reward.
>
> Applying PAGAR to IRL-based IL is equivalent to letting $R$ be the solution set of IRL.
> In this context, human demonstrations can be viewed as a supervision signal that help us identify $r$ that satisfies the conditions in Section 4.2 which are extensions of the conditions in Section 4.1 for a setting with expert demonstrations.
>
>
> > ## construction of the $R_E$ set
>
> We do not directly construct this set; rather, we utilize the aforementioned conditions to restrict the exploration of $r$.
> Our definition of $R_{E, \delta}$ is that for every $r\in R_{E,\delta}$, the difference between the optimal utility and the expert demonstration utility is within $\delta$.
> This difference equals the negative IRL loss under $r$.
> We can enforce selecting $r$ from $R_{E,\delta}$ by restricting the inverse RL loss of $r$ to be no greater than $-\delta$.
>
> > ## Example 1 ... parameterize $R_E$
>
> We mention in the paragraph above Example 1 that $\delta$ cannot be larger than the negative minimal inverse RL loss.
> When $\delta$ reaches its maximum, $R_{E,\delta}$ equals the optimal solution set of inverse RL.
> In this case, we simplify the notation as $R_E$.
> We then use Example 1 to show that in some instances, inverse RL treats the entire reward space $R$ as its optimal solution set, in other words, $R_E=R$.
> The basis function is picked to illustrate the misalignment problem in inverse RL, as the choice of the basis functions can be arbitrary.
> However, by applying PAGAR, we can identify the ground-truth reward function.
>
> > ## Is the discussion at the bottom of pg 5 meant to be a construction of $R_E$ or just a non-constructive description
>
> This discussion is about the formula in Table 1, which is an alternative of $MinimaxRegret$.
> It is equivalent to $MinimaxRegret$ in the sense that it induces the same result as $MinimaxRegret$.
> It is meant to provide more insights about PAGAR.
> It is not a constructive description.
>
>
> > ## Complexity of PAGAR
>
> In Algorithm 1, we show that PAGAR involves optimizing two policies (protagonist and antagonist) and one reward function.
> Algorithm 1 leverages existing inverse RL algorithm to perform the reward optimization and the PPO algorithm for the policy optimization.
> Therefore, the complexity of PAGAR equals the complexity of the reward optimization algorithm plus the complexity of the PPO algorithm.
> In our experiments, we use the same reward optimization algorithms as those of GAIL and VAIL in PAGAR.
>
> > ## limited number of demonstrations (10)
>
> The effective utilization of a limited number of demonstrations is widely recognized as a crucial metric for evaluating Inverse Reinforcement Learning (IRL) and Imitation Learning (IL) algorithms in the literature. It is often used to asess the performance of these algorithms, as demonstrated in notable works such as the GAIL paper [Ho. et al., 2016], the AIRL paper [Fu. et al., 2018], and various other references cited in our study.
> In GAIL [Ho. et al., 2016], which is a seminal work in imitation learning with generative adversarial networks, the algorithm demonstrates the ability to generate high-performance policies with just a single demonstration.
>
> > ## More demonstrations would reduce misalignment
>
> As we mentioned in our introduction, a reward function aligning with the expert demonstrations does not imply aligning with the underlying task.
> These two concepts are orthogonal.
> Notice that the definition of Task-Reward Alignment (Definition 1) is independent of demonstrations.
> More demonstrations may help reduce reward ambiguity (i.e. multiple demonstration-aligned rewards), which, however, cannot be eliminated even with an infinite number of data [Ng & Russell (2000); Cao et al. (2021);
> Skalse et al. (2022a;b); Skalse & Abate (2022)].

---

### Meta-Review · Area_Chair_Hf4h · 2023-12-09

**Metareview:**

This paper focuses on the reward misalignment problem in IRL, i.e. reward maximization does not guarantee task success. The proposed method  PAGAR introduces an adversarial learning setting. The overall ideas are quite novel and interesting. PAGAR is shown to improve performance on partially observable navigation environments. However, several concerns have been raised include issues with clarity of writing, and limited evaluation results. Both concerns have been raised by two reviewers. While the authors have addressed some concerns with writing in the rebuttal, the paper feels a little rushed and would do with a rewrite with clarity in mind. Further clarifying the motivation of the work in the paper would be important. I would also recommend the authors to revise the paper with clarifications during the review process. This is particularly important for ICLR as the reviewers and AC often look forward to updated discussion in the paper.

**Justification For Why Not Higher Score:**

Poorly written paper with interesting ideas. Paper needs a rewrite for the ICLR audience to understand it properly.

**Justification For Why Not Lower Score:**

N/A

---

### Decision · Program_Chairs · 2024-01-16

Reject